# FEW-SHOT IN-CONTEXT PREFERENCE LEARNING US-ING LARGE LANGUAGE MODELS

## ABSTRACT

Designing reward functions is a core component of reinforcement learning but can be challenging for truly complex behavior. Reinforcement Learning from Human Feedback (RLHF) has been used to alleviate this challenge by replacing a hand-coded reward function with a reward function learned from preferences. However, it can be exceedingly inefficient to learn these rewards as they are often learned tabula rasa. We investigate whether Large Language Models (LLMs) can reduce this query inefficiency by converting an iterative series of human preferences into code representing the rewards. We propose In-Context Preference Learning (ICPL), a method that uses the grounding of an LLM to accelerate learning reward functions from preferences. ICPL takes the environment context and task description, synthesizes a set of reward functions, and then repeatedly updates the reward functions using human feedback over videos of the resultant policies over a small number of trials. Using synthetic preferences, we demonstrate that ICPL is orders of magnitude more efficient than RLHF and is even competitive with methods that use ground-truth reward functions instead of preferences. Finally, we perform a series of human preference-learning trials and observe that ICPL extends beyond synthetic settings and can work effectively with humans-in-the-loop.

## 1 INTRODUCTION

Reward functions are a critical component of reinforcement learning (RL). However, specifying these functions becomes increasingly challenging as the complexity of the desired tasks grows. Recent advancements in pretrained foundation models have inspired approaches that leverage large language models to synthesize reward functions from task descriptions (Yu et al., 2023a; Ma et al., 2024; Yu et al., 2023b). Despite these innovations, existing methods still depend on human-designed sparse rewards or task-specific metrics to construct the reward functions. This is challenging for tasks where we cannot define any clear reward signals as the task is primarily semantically defined. For example, it is tricky to write down a reward function for a humanoid robot that corresponds to "moving like a human".

Preference-based RL offers a potential solution to this problem. Instead of relying on a human to write the reward function, we learn a reward model based on human preferences across different trajectories. This interactive approach has shown success in various RL tasks, including standard benchmarks (Christiano et al., 2017; Ibarz et al., 2018), encouraging novel behaviors (Liu et al., 2020; Wu et al., 2021), and overcoming reward exploitation (Lee et al., 2021a). However, in more complex tasks requiring extensive agent-environment interactions, preference-based RL often necessitates hundreds or even thousands of human queries to provide effective feedback. This is likely because the reward models are typically learned tabula rasa. For instance, a robotic arm button-pushing task requires over 10k queries to learn reasonable behavior (Lee et al.), which can be a major bottleneck.

In this work, we introduce a novel method, In-Context Preference Learning (ICPL), which significantly enhances the sample efficiency of preference-based RL through LLM guidance. Our primary insight is to harness the coding capabilities of LLMs to autonomously generate reward functions, then utilize human preferences through in-context learning to refine these functions. Specifically, ICPL leverages an LLM, such as GPT-4, to generate executable, diverse reward functions based on

the task description and environment source code. We acquire preferences by evaluating the agent behaviors resulting from these reward functions, selecting the most and least preferred behaviors. The selected functions, along with historical data such as reward traces of the generated reward functions from RL training, are then fed back into the LLM to guide subsequent iterations of reward function generation. We hypothesize that as a result of its grounding in text data, ICPL will be able to improve the quality of the reward function through incorporating the preferences and the history of the generated reward functions, ensuring they align more and more closely with human preferences. Unlike evolutionary search methods like EUREKA Ma et al. (2023), there is no ground-truth reward function that the LLM can use to evaluate agent performance, and thus, success here would demonstrate that LLMs have some native preference-learning capabilities.

To study the effectiveness of ICPL, we perform experiments on a diverse set of RL tasks. For scalability, we first study tasks with synthetic preferences where a ground-truth reward function is used to assign preference labels. We observe that compared to traditional preference-based RL algorithms, ICPL achieves over a 30 times reduction in the required number of preference queries to achieve equivalent or superior performance. Moreover, ICPL achieves performance comparable to reward-generation methods that utilize a ground truth sparse reward as feedback (Ma et al., 2023). Finally, we test ICPL on a particularly challenging task, "making a humanoid jump like a real human," where designing a reward is difficult. By using real human feedback, our method successfully trained an agent capable of bending both legs and performing stable, human-like jumps, showcasing the potential of ICPL in tasks where human intuition plays a critical role.

In summary, the contributions of the paper are the following:

- We propose ICPL, an LLM-based preference learning algorithm. Over a synthetic set of preferences, we demonstrate that ICPL can iteratively output rewards that increasingly reflect preferences. Via a set of ablations, we demonstrate that this improvement is relatively monotonic, suggesting that preference learning is occurring as opposed to a random search.

- We demonstrate, via human-in-the-loop trials, that ICPL is able to work effectively with humans-in-the-loop despite significantly noisier preference labels.

- We demonstrate that ICPL sharply outperforms tabula-rasa RLHF methods and is also competitive with methods that rely on access to a ground-truth reward.

## 2  RELATED WORK

**Reward Design.** In reinforcement learning, reward design is a core challenge, as most rewards both represent a desired set of behaviors and provide enough signal for learning. The most common approach to reward design is handcrafting, which requires a large number of trials by experts (Sutton, 2018; Singh et al., 2009). Since hand-coded reward design requires extensive engineering effort, several prior works have studied modeling the reward function with precollected data. For example, Inverse Reinforcement Learning (IRL) aims to recover a reward function from expert demonstration data (Arora & Doshi, 2021; Ng et al., 2000). With advances in pretrained foundation models, some recent works have also studied using large language models or vision-language models to provide reward signals (Ma et al., 2022; Fan et al., 2022; Du et al., 2023; Karamcheti et al., 2023; Kwon et al., 2023; Wang et al., 2024; Ma et al., 2024; Holk et al., 2024). Among these approaches, EUREKA (Ma et al., 2023) is the closest to our work, instructing the LLM to generate and select novel reward functions based on environment feedback with an evolutionary framework. However, EUREKA's primary goal is to test whether LLMs can produce better reward functions than humans by leveraging human-designed sparse rewards as fitness scores to evolve reward functions. In contrast, ICPL is designed for tasks even without available sparse rewards and leverages LLM grounding to accelerate learning reward functions directly from human preferences. We note that EUREKA also has a small, preliminary investigation combining human preferences with an LLM to generate human-preferred behaviors in a single scenario. Our approach relies solely on preferences, yielding higher human-involvement efficiency. This paper is a significantly scaled-up version of that investigation as well as a methodological study of how best to incorporate prior rounds of feedback.

**Human-in-the-loop Reinforcement Learning.** Feedback from humans has been proven to be effective in training reinforcement learning agents that better match human preferences (Retzlaff et al., 2024; Mosqueira-Rey et al., 2023; Kwon et al., 2023). Previous works have investigated human

feedback in various forms, such as trajectory comparisons, preferences, demonstrations, and corrections (Wirth et al., 2017; Ng et al., 2000; Jeon et al., 2020; Peng et al., 2024). Among these various methods, preference-based RL has been successfully scaled to train large foundation models for hard tasks like dialogue, e.g. ChatGPT (Ouyang et al., 2022). In LLM-based applications, prompting is a simple way to provide human feedback in order to align LLMs with human preferences (Giray, 2023; White et al., 2023; Chen et al., 2023). Iteratively refining the prompts with feedback from the environment or human users has shown promise in improving the output of the LLM (Wu et al., 2021; Nasiriany et al., 2024). This work extends the usage of the ability to control LLM behavior via in-context prompts. We aim to utilize interactive rounds of preference feedback between the LLM and humans to guide the LLM to generate reward functions that can elicit behaviors that align with human preferences.

## 3 PROBLEM DEFINITION

Our goal is to design a reward function that can be used to train reinforcement learning agents that demonstrate human-preferred behaviors. It is usually hard to design proper reward functions in reinforcement learning that induce policies that align well with human preferences.

**Markov Decision Process with Preferences( Wirth et al. (2017))** A *Markov Decision Process with Preferences* (MDPP) is defined as a tuple $M = \langle \mathcal{S}, A, \mu, \sigma, \gamma, \rho \rangle$ where $\mathcal{S}$ denotes the state space, $A$ denotes the action space, $\mu$ is the distribution of initial states, $\sigma$ is the state transition model, $\gamma \in [0, 1)$ is the discount factor. $\rho$ is the preference relation over trajectories, i.e. $\rho(\tau_i \succ \tau_j)$ denotes the probability with which trajectory $\tau_i$ is preferred over $\tau_j$. Given a set of preferences $\zeta$, the goal in an MDPP is to find a policy $\pi^*$ that maximally complies with $\zeta$. A preference $\tau_1 \succ \tau_2$ is satisfied by $\pi$ if and only if $\Pr_\pi(\tau_1) > \Pr_\pi(\tau_2)$ where $\Pr_\pi(\tau) = \mu(s_0) \prod_{t=0}^{|\tau|} \pi(a_t|s_t)\sigma(s_{t+1}|s_t, a_t)$. This can be viewed as finding a $\pi^*$ that minimizes a preference loss $L(\pi_\zeta) = \sum_i L(\pi, \zeta_i)$, where $L(\pi, \tau_1 \succ \tau_2) = -(\Pr_\pi(\tau_1) - \Pr_\pi(\tau_2))$.

**Reward Design Problem with Preferences.** A *reward design problem with preferences (RDPP)* is a tuple $P = \langle M, \mathcal{R}, A_M, \zeta \rangle$, where $M$ is a Markov Decision Process with Preferences, $\mathcal{R}$ is the space of reward functions, $A_M(\cdot) : \mathcal{R} \to \Pi$ is a learning algorithm that outputs a policy $\pi$ that optimizes a reward $R \in \mathcal{R}$ in the MDPP. $\zeta = \{(\tau_1, \tau_2)\}$ is the set of preferences. In an RDPP, the goal is to find a reward function $R \in \mathcal{R}$ such that the policy $\pi = A_M(R)$ that optimizes $R$ maximally complies with the preference set $\zeta$. In Preference-based Reinforcement Learning, the learning algorithms usually involve multiple iterations, and the preference set $\zeta$ is constructed in every iteration by sampling trajectories from the policy or policy population.

## 4 METHOD

Our proposed method, In-Context Preference Learning (ICPL), integrates LLMs with human preferences to synthesize reward functions. The LLM receives environmental context and a task description to generate an initial set of $K$ executable reward functions. ICPL then iteratively refines these functions. In each iteration, the LLM-generated reward functions are trained within the environment, producing a set of agents; we use these agents to generate videos of their behavior. A ranking is formed over the videos, from which we retrieve the best and worst reward functions corresponding to the top and bottom videos in the ranking. These selections serve as examples of positive and negative preferences. The preferences, along with additional contextual information, such as reward traces and differences from previous good reward functions, are provided as feedback prompts to the LLM. The LLM takes in this context and is asked to generate a new set of rewards. Algo. 1 presents the pseudocode, and Fig. 1 illustrates the overall process of ICPL.

### 4.1 REWARD FUNCTION INITIALIZATION

To enable the LLM to synthesize effective reward functions, it is essential to provide task-specific information, which consists of two key components: a description of the environment, including the observation and action space, and a description of the task objectives. At each iteration, ICPL ensures that $K$ executable reward functions are generated by resampling until there are $K$ executable reward functions.

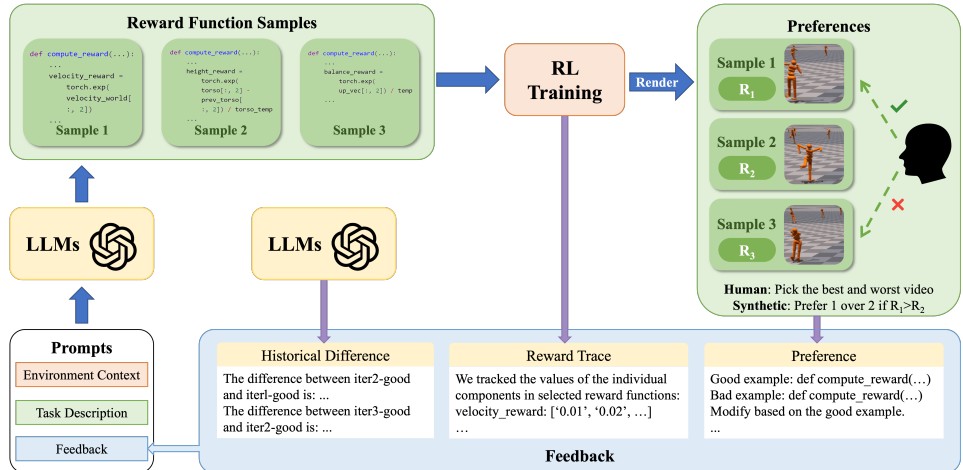

Figure 1: ICPL employs the LLM to generate initial $K$ executable reward functions based on the task description and environment context. Using RL, agents are trained with these reward functions. Videos are generated of the resultant agent behavior from which human evaluators select their most and least preferred. These selections serve as examples of positive and negative preferences. The preferences, along with additional contextual information, are provided as feedback prompts to the LLM, which is then requested to synthesize a new set of reward functions. For experiments simulating human evaluators, task scores are used to determine the best and worst reward functions.

---

**Algorithm 1:** In-Context Preference Learning (ICPL)

**Input:** Number of iterations $N$, Number of samples $K$, Environment Env, Coding LLM $\mathrm{LLM}_{RF}$

  // Initialize the prompt with environment context and task description
1   Prompt $\leftarrow$ InitializePrompt(Env)
2   **for** $i \leftarrow 1$ **to** $N$ **do**
3      $\mathrm{RF}_1, \ldots, \mathrm{RF}_K \leftarrow \mathrm{LLM}_{RF}(\text{Prompt}, K)$
      // Render videos for each reward function
4      $\mathrm{Video}_1, \ldots, \mathrm{Video}_K \leftarrow \text{Render}(\text{Env}, \mathrm{RF}_1), \ldots, \text{Render}(\text{Env}, \mathrm{RF}_K)$
      // Human selects the most preferred (G) and least preferred (B) videos
5      $G, B \leftarrow \text{Human}(\mathrm{Video}_1, \ldots, \mathrm{Video}_K)$
      // Retrieve the best and worst reward functions
6      $\text{GoodRF}, \text{BadRF} \leftarrow \mathrm{RF}_G, \mathrm{RF}_B$
      // Update the prompt with feedback
7      Prompt $\leftarrow$ GoodRF $+$ BadRF $+$ HistoricalDifference $+$ RewardTrace
8   **end**

---

## 4.2 SEARCH REWARD FUNCTIONS BY HUMAN PREFERENCES

For tasks without reward functions, the traditional preference-based approach typically involves constructing a reward model, which often demands substantial human feedback. Our approach, ICPL, aims to enhance efficiency by leveraging LLMs to directly search for optimal reward functions without the need to learn a reward model. To expedite this search process, we use an LLM-guided search to find well-performing reward functions. Specifically, we generate $K = 6$ executable reward functions per iteration across $N = 5$ iterations. In each iteration, humans select the most preferred and least preferred videos, resulting in a good reward function and a bad one. These are used as a context for the LLM to use to synthesize a new set of $K$ reward functions. These reward functions are then used in a PPO (Schulman et al., 2017) training loop, and videos are rendered of the final trained agents.

## 4.3 AUTOMATIC FEEDBACK

In each iteration, the LLM not only incorporates human preferences but also receives automatically synthesized feedback. This feedback is composed of three elements: the evaluation of selected reward functions, the differences between historical good reward functions, and the reward trace of these historical reward functions.

**Evaluation of reward functions**: The component values that make up the good and bad reward functions are obtained from the environment during training and provided to the LLM. This helps the LLM assess the usefulness of different parts of the reward function by comparing the two.

**Differences between historical reward functions**: The best reward functions selected by humans from each iteration are taken out, and for any two consecutive good reward functions, their differences are analyzed by another LLM. These differences are supplied to the primary LLM to assist in adjusting the reward function.

**Reward trace of historical reward functions**: The reward trace, consisting of the values of the good reward functions during training from all prior iterations, is provided to the LLM. This reward trace enables the LLM to evaluate how well the agent is actually able to optimize those reward components.

## 5 EXPERIMENTS

In this section, we conducted two sets of experiments to evaluate the effectiveness of our method: one using proxy human preferences and the other using real human preferences.

1) Proxy Human Preference: In this experiment, human-designed rewards, taken from EU-REKA (Ma et al., 2023), were used as proxies of human preferences. Specifically, if ground truth reward $R_1 > R_2$, sample 1 is preferred over sample 2. This method enables rapid and quantitative evaluation of our approach. It corresponds to a noise-free case that is likely easier than human trials; if ICPL performed poorly here it would be unlikely to work in human trials. Importantly, human-designed rewards were only used to automate the selection of samples and were not included in the prompts sent to the LLM; the LLM **never observes the functional form of the ground truth rewards nor does it ever receive any values from them**. Since proxy human preferences are free from noise, they offer a reliable comparison to evaluate our approach efficiently. However, as discussed later in the limitations section, these proxies may not correctly measure challenges in human feedback such as inability to rank samples, intransitive preferences, or other biases.

2) Human-in-the-loop Preference: To further validate our method, we conducted a second set of experiments with human participants. These participants repeated the tasks from the Proxy Human Preferences and engaged in an additional task that lacked a clear reward function: "Making a humanoid jump like a real human."

### 5.1 TESTBED

All experiments were conducted on tasks from the Eureka benchmark (Ma et al., 2023) based on IsaacGym, covering a diverse range of environments: *Cartpole, BallBalance, Quadcopter, Anymal, Humanoid, Ant, FrankaCabinet, ShadowHand,* and *AllegroHand.* We adhered strictly to the original task configurations, including observation space, action space, and reward computation. This ensures that our method's performance was evaluated under consistent and well-established conditions across a variety of domains.

### 5.2 BASELINES

We consider three preference-based RL methods as baselines, which update reward models during training. B-Pref (Lee et al.), a benchmark specifically designed for preference-based reinforcement learning, provides two of our baseline algorithms: **PrefPPO** and **PEBBLE**. PrefPPO is based on the on-policy RL algorithm PPO, while PEBBLE builds upon the off-policy RL algorithm SAC. Additionally, we include **SURF** (Park et al., 2022), which enhances PEBBLE by utilizing unlabeled samples with data augmentation to improve feedback efficiency. For each task, we use the default hyperparameters of PPO and SAC provided by IsaacGym, which were fine-tuned for high performance. This ensures a fair comparison across methods. Further details can be found in Appendix A.3.

### 5.3 EXPERIMENT SETUP

**Training Details.** We trained policies and rendered videos on a single A100 GPU machine. The total time for a full experiment was less than one day of wall clock time. We utilized GPT-4,

Table 1: The final task score of all methods across different tasks in IssacGym. The top result and those within one standard deviation are highlighted in bold. Standard deviations are provided in Table 6 of Appendix A.5.1 due to space limitations.

| | Cart. | Ball. | Quad. | Anymal | Ant | Human. | Franka | Shadow | Allegro |
|---|---|---|---|---|---|---|---|---|---|
| PrefPPO-49 | **499** | **499** | -1.066 | -1.861 | 0.743 | 0.457 | 0.0044 | 0.0746 | 0.0125 |
| PEBBLE-49 | **499** | **499** | -1.190 | -1.521 | 5.9891 | 0.903 | 0.0453 | 0.2142 | 0.1467 |
| SURF-49 | **499** | **499** | -1.208 | -1.35 | 0.815 | 1.675 | 0.0039 | 0.1500 | 0.1116 |
| PrefPPO-15k | **499** | **499** | -0.250 | -1.357 | 4.626 | 1.317 | 0.0399 | 0.0468 | 0.0157 |
| PEBBLE-15k | **499** | **499** | -0.231 | -0.730 | 8.543 | 4.074 | 0.6089 | 0.2438 | 0.2401 |
| SURF-15k | **499** | **499** | -0.266 | -0.346 | 7.859 | 3.292 | 0.3434 | 0.2145 | 0.2352 |
| ICPL(Ours) | **499** | **499** | **-0.0195** | **-0.007** | **12.04** | **9.227** | **0.9999** | **13.231** | **25.030** |
| Eureka | 499 | 499 | -0.023 | -0.003 | 10.86 | 9.059 | 0.9999 | 11.532 | 25.250 |

specifically the GPT-4-0613, as the backbone LLM in the Proxy Human Preference experiment. For the Human-in-the-loop Preference experiment, we employ GPT-4o.

**Evaluation Metric.** Here, we provide a specific explanation of how sparse rewards (detailed in Appendix A.4) are used as task metrics in the adopted IsaacGym tasks. The task metric is the average of the sparse rewards across parallel environments. To assess the generated reward function or the learned reward model for each RL run, we take the maximum task metric value sampled at fixed intervals, marked as *task score of reward function/model* (RTS). In each iteration, ICPL generates 6 RL runs and selects the highest RTS as the result for that iteration. ICPL performs 5 iterations and then selects the highest RTS from these iterations as the *task score* (TS) for each experiment. Due to the inherent randomness of LLMs, we run 5 experiments for all methods, and report the highest TS as the *final task score* (FTS) for each approach. A higher FTS indicates better performance across all tasks.

## 5.4 RESULTS OF PROXY HUMAN PREFERENCE

### 5.4.1 MAIN RESULTS

In ICPL, we use human-designed sparse rewards as proxies to simulate ideal human preferences. Specifically, in each iteration, we select the reward function with the highest RTS as the good example and the reward function with the lowest RTS as the bad example for feedback. All baseline methods leverage dense rewards to simulate proxy human preference, offering a stronger and more informative signal for labeling preferences. If the cumulative dense reward of trajectory 1 is greater than that of trajectory 2, then trajectory 1 is preferred over trajectory 2. We also tried sparse rewards as proxy human preference in baseline methods and observed similar performance. Table 1 shows the final task score (FTS) for all methods across IsaacGym tasks.

For ICPL and baselines, we track the number of synthetic queries $Q$ required as a proxy for measuring the likely real human effort involved, which is crucial for methods that rely on human-in-the-loop preference feedback. Specifically, we define a single query as a human comparing two trajectories and providing a preference. In ICPL, each iteration generates $K$ reward function samples, resulting in $K$ corresponding videos. The human compares these videos, first selecting the best one, then picking the worst from the remaining $K - 1$ videos. After $N = 5$ iterations, the best video of each iteration is compared to select the overall best. The number of human queries $Q$ can be calculated as $Q = (K - 1) \times 2N - 1$. For ICPL, with $K = 6$ and $N = 5$, this results in $Q = 49$. In baselines, the simulated human teacher compares two sampled trajectories and provides a preference label to update the reward model. We set the maximum number of queries to $Q = 49$, matching ICPL, and also test $Q = 15k$, denoted as Baseline-#$Q$ in Table 1, to compare the final task score (FTS) across different tasks. Additional results with $Q = 150, 1.5k$ can be found in Table 6 of Appendix A.5.1.

As shown in Table 1, for the simpler tasks like *Cartpole* and *BallBalance*, all methods achieve equal performance. Notably, we observe that for these particularly simple tasks, ICPL can generate correct reward functions in a zero-shot manner, without requiring feedback. As a result, ICPL only requires querying the human 5 times, while baseline methods, after 5 queries, fail to train a reasonable reward model with the preference-labeled data. For relatively more challenging tasks, Baseline-49 performs significantly worse than ICPL when using the same number of human queries. In fact, Baseline-49 fails in most tasks. As the number of human queries increases, baselines' performance improves

Table 2: Ablation studies on ICPL modules. The runs have fairly high variance so we highlight the top two results in bold. The full table with std. deviations included can be found in Appendix A.5.1. We observe that ICPL with all of the components is consistently the best performing, suggesting that most of the components are useful.

|  | Cart. | Ball. | Quad. | Anymal | Ant | Human. | Franka | Shadow | Allegro |
|---|---|---|---|---|---|---|---|---|---|
| ICPL w/o RT | **499** | **499** | -0.0340 | -0.387 | 10.50 | 8.337 | **0.9999** | 10.769 | **25.641** |
| ICPL w/o RTD | **499** | **499** | -0.0216 | **-0.009** | 10.53 | **9.419** | **1.0000** | 11.633 | 23.744 |
| ICPL w/o RTDB | **499** | **499** | **-0.0136** | -0.014 | **11.97** | 8.214 | 0.5129 | **13.663** | **25.386** |
| OpenLoop | **499** | **499** | -0.0410 | -0.016 | 9.350 | 8.306 | **0.9999** | 9.476 | 23.876 |
| ICPL(Ours) | **499** | **499** | **-0.0195** | **-0.007** | **12.04** | 9.227 | **0.9999** | **13.231** | 25.030 |

across most tasks, but it still falls noticeably short compared to ICPL. This demonstrates that ICPL, with the integration of LLMs, can reduce human effort in preference-based learning by at least 30 times.

**Performance Analysis with Eureka** We further report Eureka's performance (Ma et al., 2023) as an approximate upper bound on the expected performance ICPL could achieve. Eureka is an LLM-powered reward design method that uses sparse rewards as fitness scores. Specifically, the reward function with the highest RTS is selected as the candidate reward function for feedback in each iteration and RTS is incorporated as the "task score" in the reward reflection. Original Eureka generates 16 reward functions in each iteration without checking their executability, assuming at least one will typically work across all considered environments in the first iteration. To ensure a fair comparison, we modified Eureka to generate a fixed number of executable reward functions, specifically $K = 6$ per iteration, the same as ICPL. This adjustment improves Eureka's performance in more challenging tasks, where it often generates fewer executable reward functions. As shown in Table 1, ICPL surprisingly achieves comparable performance, indicating that ICPL's use of LLMs for preference learning is effective.

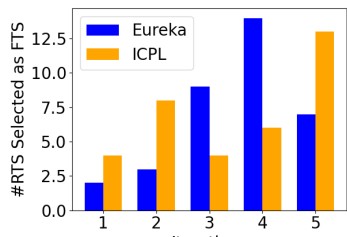

Figure 2: Distribution of which iteration is selected as the top-scoring iteration. While it is not perfectly monotonic, we observe that the final iteration is generally the best one, suggesting that the inferred reward is gradually approaching the ground-truth reward.

From the analysis conducted across 7 tasks where zero-shot generation of optimal reward functions was not feasible in the first iteration, we examined which iteration's RTS was chosen as the final FTS. The distribution of RTS selections over iterations is illustrated in Fig. 2. The results indicate that FTS selections do not always come from the last iteration; some are also derived from earlier iterations. However, the majority of FTS selections originate from iterations 4 and 5, suggesting that ICPL is progressively refining and enhancing the reward functions over successive iterations as opposed to randomly generating diverse reward functions.

## 5.5 METHOD ANALYSIS

To validate the effectiveness of ICPL's module design, we conducted ablation studies. We aim to answer several questions that could undermine the results presented here:

1. Are components such as the reward trace or the reward difference helpful?
2. Is the LLM actually performing preference learning? Or is it simply zero-shot outputting the correct reward function due to the task being in the training data?

### 5.5.1 ABLATIONS

The results of the ablations are shown in Table 2. In these studies, "ICPL w/o RT" refers to removing the reward trace from the prompts sent to the LLMs. "ICPL w/o RTD" indicates the removal of both the reward trace and the differences between historical reward functions from the prompts. "ICPL w/o RTDB" removes the reward trace, differences between historical reward functions, and bad reward functions, leaving only the good reward functions and their evaluation in the prompts. The "OpenLoop" configuration samples $K \times N$ reward functions without any feedback, corresponding to the ability of the LLM to zero-shot accomplish the task.

Due to the large variance of the experiments (see Appendix), we mark the top two results in bold. As shown, ICPL achieves top 2 results in 8 out of 9 tasks and is comparable on the *Allegro* task. The "OpenLoop" configuration performs the worst, indicating that our method does not solely rely on GPT-4's either having randomly produced the right reward function or having memorized the reward function during its training. This improvement is further demonstrated in Sec. 5.5.2, where we show the step-by-step improvements of ICPL through proxy human preference feedback. Additionally, "ICPL w/o RT" underperforms on multiple tasks, highlighting the importance of incorporating the reward trace of historical reward functions into the prompts.

### 5.5.2 IMPROVEMENT ANALYSIS

Table 1 presents the performance achieved by ICPL. While it is possible that the LLMs could generate an optimal reward function in a zero-shot manner, the primary focus of our analysis is not solely on absolute performance values. Rather, we emphasize whether ICPL is capable of enhancing performance through the iterative incorporation of preferences. We calculated the average RTS improvement over iterations relative to the first iteration for the two tasks with the largest improvements compared with "OpenLoop", *Ant* and *ShadowHand*. As shown in Fig. 3, the RTS exhibits an upward trend, demonstrating its effectiveness in improving reward functions over time. We note that this trend is roughly monotonic, indicating that on average the LLM is using the preferences to construct reward functions that are closer to the ground-truth reward. We further use an example in the *Humanoid* task to demonstrate how ICPL progressively generated improved reward functions over successive iterations in Appendix A.5.2.

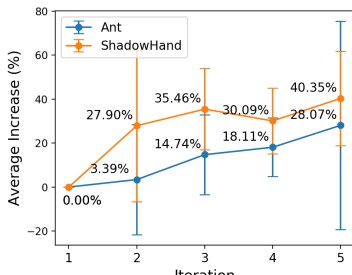

Figure 3: Average improvement of the Reward Task Score (RTS) over successive iterations relative to the first iteration in ICPL for the Ant and ShadowHand tasks, demonstrating the method's effectiveness in refining reward functions over time.

### 5.6 RESULTS OF HUMAN-IN-THE-LOOP PREFERENCE

To address the limitations of proxy human preferences, which simulate idealized human preference and may not fully capture the challenges humans may face in providing preferences, we conducted experiments with real human participants. We recruited 7 volunteers for human-in-the-loop experiments, with 5 assigned to IsaacGym tasks and 2 to a newly designed task. Additionally, 20 volunteers were recruited to evaluate the performance of different methods. None of the volunteers had prior experience with these tasks, ensuring an unbiased evaluation based on their preferences.

### 5.6.1 HUMAN EXPERIMENT SETUP

Before the experiment, each volunteer was provided with a detailed explanation of the experiment's purpose and process. Additionally, volunteers were fully informed of their rights, and written consent was obtained from each participant. The experimental procedure was approved by the department's ethics committee to ensure compliance with institutional guidelines on human subject research.

In ICPL experiments, each volunteer was assigned an account with a pre-configured environment to ensure smooth operation. After starting the experiment, LLMs generated the first iteration of reward functions. Once the reinforcement learning training was completed, videos corresponding to the policies derived from each reward function were automatically rendered. Volunteers compared the behaviors in the videos with the task descriptions and selected both the best and the worst-performing videos. They then entered the respective identifiers of these videos into the interactive interface and pressed "Enter" to proceed. The human preference was processed as an LLM prompt for generating feedback, leading to the next iteration of reward function generation.

This training-rendering-selection process was repeated across several iterations. At the end of the final iteration, the volunteers were asked to select the best video from those previously marked as good, designating it as the final result of the experiment. For IsaacGym tasks, the corresponding RTS was recorded as TS. It is important to note that, unlike proxy human preference experiments where the TS is the maximum RTS across iterations, in the human-in-the-loop preference experiment, TS refers to the highest RTS chosen by the human, as human selections are not always based on the

Table 3: The final task score of human-in-the-loop preference across 5 IsaacGym tasks. The values in parentheses represent the standard deviation.

|           | Quadcopter      | Ant           | Humanoid     | Shadow        | Allegro        |
|-----------|-----------------|---------------|--------------|---------------|----------------|
| OpenLoop  | -0.0410(0.32)   | 9.350(2.35)   | 8.306(1.63)  | 9.476(2.44)   | 23.876(7.91)   |
| ICPL-proxy| -0.0195(0.09)   | 12.040(1.69)  | 9.227(0.93)  | 13.231(1.88)  | 25.030(3.72)   |
| ICPL-real | -0.0183(0.29)   | 11.142(0.37)  | 8.392(0.53)  | 10.74(0.92)   | 24.134 (6.52)  |

maximum RTS at each iteration. Given that ICPL required reinforcement learning training in every iteration, each experiment lasted two to three days. Each volunteer was assigned a specific task and conducted five experiments, one for each task, with the highest TS being recorded as FTS in IsaacGym tasks.

### 5.6.2 ISAACGYM TASKS

Due to the simplicity of the *Cartpole*, *BallBalance*, *Franka* tasks, where LLMs were able to zero-shot generate correct reward functions without any feedback, these tasks were excluded from the human trials. The *Anymal* task, which involved commanding a robotic dog to follow random commands, was also excluded as it was difficult for humans to evaluate whether the commands were followed based solely on the videos. For the 5 adopted tasks, we describe in the Appendix A.6.2 how humans infer tasks through videos and the potential reasons that may lead to preference rankings that do not accurately reflect the task.

Table 3 presents the FTS for the human-in-the-loop preference experiments conducted across 5 suitable IsaacGym tasks, labeled as "ICPL-real". The results of the proxy human preference experiment are labeled as "ICPL-proxy". As observed, the performance of "ICPL-real" is comparable or slightly lower than that of "ICPL-proxy" in all 5 tasks, yet it still outperforms the "OpenLoop" results in 3 out of 5 tasks. This indicates that while humans may have difficulty providing consistent preferences from videos as proxies, their feedback can still be effective in improving performance when combined with LLMs.

### 5.6.3 HUMANOIDJUMP TASK

In our study, we introduced a new task: *HumanoidJump*, with the task description being "to make humanoid jump like a real human." Defining a precise task metric for this objective is challenging, as the criteria for human-like jumping are not easily quantifiable. The task-specific prompts used in this experiment are detailed in the Appendix A.6.3.

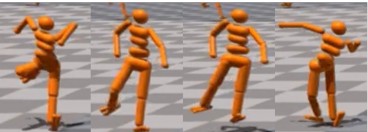

Figure 4: A common behavior.

The most common behavior observed in this task, as illustrated in Fig. 4, is what we refer to as the "leg-lift jump." This behavior involves initially lifting one leg to raise the center of mass, followed by the opposite leg pushing off the ground to achieve lift. The previously lifted leg is then lowered to extend airtime. Various adjustments of the center of mass with the lifted leg were also noted. This behavior meets the minimal metric of a jump: achieving a certain distance off the ground. If feedback were provided based solely on this minimal metric, the "leg-lift jump" would likely be selected as a candidate reward function. However, such candidates show limited improvement in subsequent iterations, failing to evolve into more human-like jumping behaviors.

Conversely, when real human preferences were used to guide the task, the results were notably different. The volunteer judged the overall quality of the humanoid's jump behavior instead of just the metric of leaving the ground. Fig. 5 illustrates an example where the volunteer successfully guided the humanoid towards a more human-like jump by selecting behaviors that, while initially not optimal, displayed promising movement patterns. The reward functions are shown in Appendix A.6.3. In the first iteration, "leg-lift jump" was not selected despite the humanoid jumping off the ground. Instead, a video where the humanoid appears to attempt a jump using both legs, without leaving the ground, was chosen. By the fifth and sixth iterations, the humanoid demonstrated more sophisticated behaviors, such as bending both legs and lowering the upper body to shift the center of mass, behaviors that are much more akin to a real human jump.

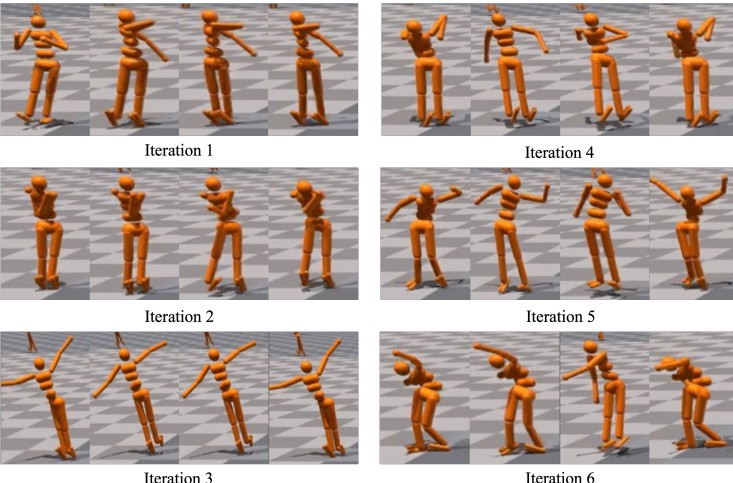

Figure 5: The humanoid learns a human-like jump by bending both legs and lowering the upper body to shift the center of mass in a trial of human-in-the-loop experiments. Note that both legs are used to jump and the agent bends at the hips.

**Quantitative Evaluation.** We conducted additional experiments using the "OpenLoop" configuration, which generates $K \times N$ reward functions without any feedback, on the HumanoidJump task. In this configuration, we performed 5 independent experiments, each comprising 6 iterations with 6 samples per iteration. A volunteer selected the most preferred video as the final result. For quantitative evaluation, 20 additional volunteers were recruited to compare the performance of ICPL and OpenLoop. Each volunteer indicated their preference between two videos presented in random order—one generated by ICPL and the other by OpenLoop. The results showed that 17 out of 20 participants preferred the ICPL agent, demonstrating that ICPL produces behaviors more aligned with human preferences.

| Method | Vote |
|---|---|
| OpenLoop | 3/20 |
| ICPL | 17/20 |

Table 4: Human Preferences

## 6 CONCLUSION

Our proposed method, In-Context Preference Learning (ICPL), demonstrates significant potential for addressing the challenges of preference learning tasks through the integration of large language models. By leveraging the generative capabilities of LLMs to autonomously produce reward functions, and iteratively refining them using human feedback, ICPL reduces the complexity and human effort typically associated with preference-based RL. Our experimental results, both in proxy human and human-in-the-loop settings, show that ICPL not only surpasses traditional RLHF in efficiency but also competes effectively with methods utilizing ground-truth rewards instead of preferences. Furthermore, the success of ICPL in complex, subjective tasks like humanoid jumping highlights its versatility in capturing nuanced human intentions, opening new possibilities for future applications in complex real-world scenarios where traditional reward functions are difficult to define.

**Limitations.** While ICPL demonstrates significant potential, it faces limitations in tasks where human evaluators struggle to assess performance from video alone, such as *Anymal*'s "follow random commands." In such cases, subjective human preferences may not provide adequate guidance. Future work will explore integrating human preferences with artificially designed metrics to enhance the ease with which humans can assess the videos, ensuring more reliable performance in complex tasks. Additionally, we observe that the performance of the task is qualitatively dependent on the diversity of the initial reward functions that seed the search. While we do not study methods to achieve this here, relying on the LLM to provide this initial diversity is a current limitation. Furthermore, the limited number of participants in human-in-the-loop experiments may restrict the generalizability of our findings, as it might not fully capture the broad range of human preferences. Another limitation of ICPL is that each iteration involves training new RL policies, resulting in a waiting period of several hours for participants before they can provide additional feedback. This could be addressed by continuously training an RL agent under non-stationary reward functions, which presents a promising direction for future work.

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

# A   APPENDIX

We would suggest visiting `https://sites.google.com/view/few-shot-icpl/home` for more information and videos.

## A.1   FULL PROMPTS

The prompts used in ICPL for synthesizing reward functions are presented in Prompts 1, 2, and 3. The prompt for generating the differences between various reward functions is shown in Prompt 4.

### Prompt 1: Initial System Prompts of Synthesizing Reward Functions

```
You are a reward engineer trying to write reward functions to solve reinforcement learning
    tasks as effective as possible.
Your goal is to write a reward function for the environment that will help the agent learn the
    task described in text.
Your reward function should use useful variables from the environment as inputs. As an example
    , the reward function signature can be:
@torch.jit.script
def compute_reward(object_pos: torch.Tensor, goal_pos: torch.Tensor) -> Tuple[torch.Tensor,
    Dict[str, torch.Tensor]]:
    ...
    return reward, {}
Since the reward function will be decorated with @torch.jit.script, please make sure that the
    code is compatible with TorchScript (e.g., use torch tensor instead of numpy array).
Make sure any new tensor or variable you introduce is on the same device as the input tensors.
```

### Prompt 2: Feedback Prompts

```
The reward function has been iterated {current_iteration} rounds.
In each iteration, a good reward function and a bad reward function are generated.
The good reward function generated in the x-th iteration is denoted as "iterx-good", and the
    bad reward function generated is denoted as "iterx-bad".
The following outlines the differences between these reward functions.

We trained an RL policy using iter1-good reward function code and tracked the values of the
    individual components in the reward function after every {epoch_freq} epochs and the
    maximum, mean, minimum values encountered:
<REWARD FEEDBACK>

The difference between iter2-good and iter1-good is: <DIFFERENCE>

<REPEAT UNTIL THE CURRENT ITERATION>

Next, the two reward functions generated in the {current_iteration_ordinal} iteration are
    provided.
The 1st generated reward function is as follows:
<REWARD FUNCTION>
We trained an RL policy using the 1st reward function code and tracked the values of the
    individual components in the reward function after every {epoch_freq} epochs and the
    maximum, mean, minimum values encountered:
<REWARD FEEDBACK>

The 2nd generated reward function is as follows:
<REWARD FUNCTION>
We trained an RL policy using the 2nd reward function code and tracked the values of the
    individual components in the reward function after every {epoch_freq} epochs and the
    maximum, mean, minimum values encountered:
<REWARD FEEDBACK>

The following content is the most important information.
Good example: 1st reward function. Bad example: 2nd reward function.
You need to modify based on the good example. DO NOT based on the code of the bad example.
Please carefully analyze the policy feedback and provide a new, improved reward function that
    can better solve the task. Some helpful tips for analyzing the policy feedback:
    (1) If the values for a certain reward component are near identical throughout, then this
    means RL is not able to optimize this component as it is written. You may consider
        (a) Changing its scale or the value of its temperature parameter
        (b) Re-writing the reward component
        (c) Discarding the reward component
    (2) If some reward components' magnitude is significantly larger, then you must re-scale
    its value to a proper range
Please analyze each existing reward component in the suggested manner above first, and then
    write the reward function code.
```

Prompt 3: Prompts of Tips for Writing Reward Functions

```
The output of the reward function should consist of two items:
    (1) the total reward,
    (2) a dictionary of each individual reward component.
The code output should be formatted as a python code string: "```python ... ```".

Some helpful tips for writing the reward function code:
    (1) You may find it helpful to normalize the reward to a fixed range by applying
    transformations like torch.exp to the overall reward or its components
    (2) If you choose to transform a reward component, then you must also introduce a
    temperature parameter inside the transformation function; this parameter must be a named
    variable in the reward function and it must not be an input variable. Each transformed
    reward component should have its own temperature variable
    (3) Make sure the type of each input variable is correctly specified; a float input
    variable should not be specified as torch.Tensor
    (4) Most importantly, the reward code's input variables must contain only attributes of
    the provided environment class definition (namely, variables that have prefix self.).
    Under no circumstance can you introduce new input variables.
```

Prompt 4: Prompts of Describing Differences

```
You are an engineer skilled at comparing the differences between two reward function code
    snippets used in reinforcement learning.
Your goal is to describe the differences between two reward function code snippets.
The following are two reward functions written in Python code used for the task:
<TASK_DESCRIPTION>
The first reward function is as follows:
<REWARD_FUNCTION>
The second reward function is as follows:
<REWARD_FUNCTION>
Please directly describe the differences between these two codes. No additional descriptions
    other than the differences are required.
```

## A.2 ICPL DETAILS

The full pseudocode of ICPL is listed in Algo. 2.

## A.3 BASELINE DETAILS

### A.3.1 PREFPPO

The baseline PrefPPO adopted in our experiments comprises two primary components: agent learning and reward learning, as outlined in Lee et al. (2021c). Algo. 3 illustrates the pseudocode for PrefPPO. Throughout this process, the method maintains a policy denoted as $\pi_\varphi$ and a reward model represented by $\hat{r_\psi}$.

**Agent Learning**. In the agent learning phase, the agent interacts with the environment and collects experiences. The policy is subsequently trained using reinforcement learning, to maximize the cumulative rewards provided by the reward model $\hat{r_\psi}$. We utilize the on-policy reinforcement learning algorithm PPO (Schulman et al., 2017) as the backbone algorithm for training the policy. Additionally, we apply unsupervised pre-training to match the performance of the original benchmark. Specifically, during earlier iterations, when the reward model has not collected sufficient trajectories and exhibits limited progress, we utilize the state entropy of the observations, defined as $H(s) = -\mathbb{E}_{s \sim p(s)}[\log p(s)]$, as the goal for agent training. During this process, trajectories of varying lengths are collected. Formally, a trajectory $\sigma$ is defined as a sequence of observations and actions $(s_1, a_1), \ldots, (s_t, a_t)$ that represents the complete interaction of the agent with the environment, concluding at timestep $t$.

**Reward Learning**. A preference predictor is developed using the current reward model to align with human preferences, formulated as follows:

$$P_\psi[\sigma^1 \succ \sigma^0] = \frac{\exp\left(\sum_t \hat{r_\psi}(s_t^1, a_t^1)\right)}{\sum_{i \in \{0,1\}} \exp\left(\sum_t \hat{r_\psi}(s_t^i, a_t^i)\right)}, \quad (1)$$

where $\sigma_0 = (s_1^0, a_1^0), \ldots, (s_{l_0}^0, a_{l_0}^0)$ and $\sigma_1 = (s_1^1, a_1^1), \ldots, (s_{l_1}^1, a_{l_1}^1)$ represent two complete trajectories with different trajectory length $l_0$ and $l_1$. $P_\psi[\sigma^1 \succ \sigma^0]$ denotes the probability that trajectory

---

**Algorithm 2:** ICPL

---

**Input:** # iterations $N$, # samples in each iterations $K$, environment Env, coding LLM $\text{LLM}_{RF}$, difference LLM $\text{LLM}_{Diff}$

1 **Function** Feedback(Env, RF)**:**
2    **return** The values of each component that make up RF during the training process in Env
3 **Function** History(RFlist, Env, $\text{LLM}_{Diff}$)**:**
4    HistoryFeedback $\leftarrow$ ""
5    **for** $i \leftarrow 1$ *to* ***len***(RFlist) $- 1$ **do**
       // The reward trace of historical reward functions
6        HistoryFeedback $\leftarrow$ HistoryFeedback $+$ Feedback(Env, RFlist$[i-1]$)
       // The differences between historical reward functions
7        HistoryFeedback $\leftarrow$
         HistoryFeedback $+ \text{LLM}_{Diff}$(DifferencePrompt $+$ RFlist$[i]$ $+$ RFlist$[i-1]$)
8    **end**
9    **return** HistoryFeedback
   // Initialize the prompt containing the environment context and task description
10 Prompt $\leftarrow$ InitializePrompt
11 RFlist $\leftarrow$ []
12 **for** $i \leftarrow 1$ *to* $N$ **do**
13    $\text{RF}_1, \ldots, \text{RF}_K \leftarrow \text{LLM}_{RF}$(Prompt, $K$)
14    **while** *any of* $\text{RF}_1, \ldots, \text{RF}_K$ *is not executable* **do**
15        $j_1, \ldots, j_{K'} \leftarrow$ Index of non-executable reward functions
       // Regenerate non-executable reward functions
16        $\text{RF}_{j_1}, \ldots, \text{RF}_{j'_K} \leftarrow \text{LLM}_{RF}$(Prompt, $K'$)
17    **end**
   // Render videos for sampled reward functions
18    $\text{Video}_1, \ldots, \text{Video}_K \leftarrow$ Render(Env, $\text{RF}_1$)$, \ldots,$ Render(Env, $\text{RF}_K$)
   // Human selects the most preferred and least preferred videos
19    $G, B \leftarrow$ Human($\text{Video}_1, \ldots, \text{Video}_K$)
20    GoodRF, BadRF $\leftarrow \text{RF}_G, \text{RF}_B$
21    RFlist.**append**(GoodRF)
   // Update prompt for feedback
22    Prompt $\leftarrow$
     GoodRF $+$ Feedback(Env, GoodRF) $+$ BadRF $+$ Feedback(Env, BadRF) $+$ PreferencePrompt
23    Prompt $\leftarrow$ Prompt $+$ History(RFlist, Env, $\text{LLM}_{Diff}$)
24 **end**

---

$\sigma^1$ is preferred over $\sigma^0$ as indicated by the preference predictor. In the original PrefPPO framework, test task trajectories are of fixed length, allowing for the extraction of fixed-length segments to train the reward model. However, the tasks in this paper have varying trajectory lengths, so we use full trajectory pairs as training data instead of segments. We also tried zero-padding trajectories to the maximum episode length and then segmenting them, but this approach was ineffective in practice.

To provide more effective labels, the original PrefPPO utilizes dense rewards $r$ to simulate oracle human preferences, which is

$$P[\sigma^1 \succ \sigma^0] = \begin{cases} 1 & \text{If } \sum_t r(s_t^1, a_t^1) > \sum_t r(s_t^1, a_t^1) \\ 0 & \text{Otherwise} \end{cases}. \qquad (2)$$

The probability $P[\sigma^1 \succ \sigma^0]$ reflects the preference of the ideal teacher, which is perfectly rational and deterministic, without incorporating noise. We utilize the default dense rewards in the adopted IsaacGym tasks, which differ from ICPL and EUREKA, both of which use sparse rewards (task metrics) as the proxy preference. While we also experimented with sparse rewards in PrefPPO and found similar performance (refer to Table 8), we opted to retain the original PrefPPO approach in all experiments. The reward model is trained by minimizing the cross-entropy loss between the predictor and labels, utilizing trajectories sampled from the agent learning process. Note that since the agent learning process requires significantly more experiences for training than reward training, we only use trajectories from a subset of the environments for reward training.

To sample trajectories for reward learning, we employ the disagreement sampling scheme from Lee et al. (2021c) to enhance the training process. This scheme first generates a larger batch of trajectory pairs uniformly at random and then selects a smaller batch with high variance across an ensemble of preference predictors. The selected pairs are used to update the reward model.

For a fair comparison, we recorded the number of times PrefPPO queried the oracle human simulator to compare two trajectories and obtain labels during the reward learning process, using this as a measure of the human effort involved. In the proxy human experiment, we set the maximum number of human queries $Q$ to $49, 150, 1.5k, 15k$. Once this limit is reached, the reward model ceases to update, and only the policy model is updated via PPO. Algo. 4 illustrates the pseudocode for reward learning.

### A.3.2   PEBBLE

PEBBLE (Lee et al., 2021b) is a popular feedback-efficient preference-based RL algorithm. It improves the feedback efficiency of the algorithm by mainly utilizing two modules: unsupervised pre-training and off-policy learning. The unsupervised pre-training module is introduced in the PrefPPO section, and we also include it in PEBBLE with the same setting. PEBBLE utilizes the off-policy algorithm SAC (Haarnoja et al., 2018) instead of PPO as the backbone RL algorithm. SAC stores the agent's past experiences in a replay buffer and reuses these experiences during the training process. PEBBLE relabels all past experiences in the replay buffer every time it updates the reward model.

### A.3.3   SURF

SURF (Park et al., 2022) is a framework that uses unlabeled samples with data augmentation to improve the efficiency of reward training. In our experiments, the length of trajectories is varied and may affect the evaluation of the trajectories. Therefore, we do not apply the data augmentation technique and only utilize the semi-supervised learning method in SURF.

In addition to the labeled pairs of trajectories $\mathcal{D}_l = \{(\sigma_l^0, \sigma_l^1, y)^i\}_{i=1}^{N_l}$, SURF samples another unlabeled dataset $\mathcal{D}_U = \{(\sigma_u^0, \sigma_u^1)^i\}_{u=1}^{N_u}$ to optimize the reward model. Specifically, during each update of the reward model, SURF not only samples a set of trajectories and queries a human teacher for labels, but also samples additional trajectory pairs. These additional pairs are assigned pseudo-labels generated by the current reward model.

$$\hat{y}_u(\sigma_u^0, \sigma_u^1) = \begin{cases} 1 & \text{If } P_\psi[\sigma_u^1 \succ \sigma_u^0] > 0.5. \\ 0 & \text{Otherwise}. \end{cases} \qquad (3)$$

Here $\psi$ is the preference predictor based on the current reward model. During the training process of reward model, SURF will also use the unlabeled samples for training if the confidence of the

predictor is higher than a pre-defined threshold. In experiments, we follows the implementation of SURF (Park et al., 2022).

---

**Algorithm 3:** PrefPPO

---

**Input:** # iterations $B$, # unsupervised learning iterations $M$, # rollout steps $S$, reward model $\hat{r_\psi}$, # environments for reward learning $E$, # iterations for collecting trajectories RewardTrainingInterval, maximal number of human queries $Q$, environments Env

1 HumanQueryCount $\leftarrow 0$
2 Trajectories $\leftarrow []$
3 **Function** TrainReward($\hat{r_\psi}$, Trajectories):
4 **Function** CollectRollout(RewardType, $S$, Policy, $\hat{r_\psi}$, Env):
5    RolloutBuffer $\leftarrow []$
6    **for** $j \leftarrow 1$ *to* $S$ **do**
7       Action $\leftarrow$ Policy(Observation)
      // Here EnvDones is a binary sequence replied from the envrioment,
          representing whether the environments are done.
8       NewObservation, EnvReward, EnvDones $\leftarrow$ Env(Actions)
9       **if** RewardType == Unsuper **then**
10          PredReward $\leftarrow$ ComputeStateEntropy(Observation)
11       **end**
12       **else**
13          PredReward $\leftarrow \hat{r_\psi}$(Observation, Action)
14       **end**
      // Collect trajectories for reward learning
15       Trajectories $\leftarrow$ Trajectories $+$ (Observation, Action, EnvDones, EnvReward)
      // Add complete trajectory to reward model
16       **for** $k \leftarrow 1$ *to* $E$ **do**
17          **if** EnvDones[Env[$k$]] **then**
18             AddTrajectory($\hat{r_\psi}$, Trajectories[$k$])
19             Trajectories[$k$] $\leftarrow []$
20          **end**
21       **end**
      // Reward Learning
22       **if** $j$ is divisible by RewardTrainingInterval *and* HumanQueryCount $< Q$ **then**
23          $\hat{r_\psi} \leftarrow$ TrainReward($\hat{r_\psi}$, Trajectories)
24       **end**
      // Collect rollouts for agent learning
25       RolloutBuffer $\leftarrow$ RolloutBuffer $+$ (Observation, Action, PredReward)
26       Observation $\leftarrow$ NewObservation
27    **end**
28    **return** RolloutBuffer
29 Policy $\leftarrow$ Initialize
30 **for** $i \leftarrow 1$ *to* $B$ **do**
   // Collect rollouts and trajectories
31    **if** $i < M$ **then**
32       RolloutBuffer $\leftarrow$ CollectRollout(Unsuper, $S$, Policy, $\hat{r_\psi}$, Env)
33    **end**
34    **else**
35       RolloutBuffer $\leftarrow$ CollectRollout(RewardModel, $S$, Policy, $\hat{r_\psi}$, Env)
36    **end**
   // Agent Learning: Train agent with the collect RolloutBuffer via PPO, omitted here
37    AgentLearning(Policy, RolloutBuffer)
38 **end**

---

---

**Algorithm 4:** Reward Learning of PrefPPO

---

**Input:** reward model $\hat{r_\psi}$, # samples for human queries per time `MbSize`, # maximal iterations for reward learning `MaxUpdate`, maximal number of human queries $Q$, environments `Env`

1   `LabeledQueries` $\leftarrow$ []
2   `HumanQueryCount` $\leftarrow$ 0
3   **Function** `TrainReward`($\hat{r_\psi}$, `Trajectories`)**:**
      // Use disagreement sampling to sample trajectories
4      $\sigma_0, \sigma_1 \leftarrow$ `DisagreementSampling`(`Trajectories`, `MbSize`)
5      **for** $(x_0, x_1)$ *in* $(\sigma_0, \sigma_1)$ **do**
         // Give oracle human preferences between two trajectories according to the sum of dense reward.
6         `LabeledQueries` $\leftarrow$ `LabeledQueries` + $(x_0, x_1,$ `HumanQuery`$(x_0, x_1))$
         // In experiments, we do not add HumanQueryCount if the pair has already been queried before
7         `HumanQueryCount` $\leftarrow$ `HumanQueryCount` $+ 1$
8         **if** `HumanQueryCount` $> Q$ **then**
9            |   BREAK
10        **end**
11     **end**
12     **for** $i \leftarrow 1$ *to* `MaxUpdate` **do**
         // Update reward model by minimizing the cross entropy loss and record the accuracy on all pairs.
13        $\hat{r_\psi}$, `Accuracy` $\leftarrow$ `RewardLearning`($\hat{r_\psi}$, `LabeledQueries`)
14        **if** `Accuracy` $\geq 97\%$ **then**
15          |   BREAK
16        **end**
17     **end**
18     **return** $\hat{r_\psi}$

---

### A.4   ENVIRONMENT DETAILS

In Table 5, we present the observation and action dimensions, along with the task description and task metrics for 9 tasks in IsaacGym.

### A.5   PROXY HUMAN PREFERENCE

#### A.5.1   ADDITIONAL RESULTS

Due to the high variance in LLMs performance, we report the standard deviation across 5 experiments as a supplement, which is presented in Table 6 and Table 7. We also report the final task score of PrefPPO using sparse rewards as the preference metric for the simulated teacher in Table 8.

#### A.5.2   IMPROVEMENT ANALYSIS

We use a trial of the *Humanoid* task to illustrate how ICPL progressively generated improved reward functions over successive iterations. The task description is "to make the humanoid run as fast as possible". Throughout five iterations, adjustments were made to the penalty terms and reward weightings. In the first iteration, the total reward was calculated as $0.5 \times$ speed_reward $+ 0.25 \times$ deviation_reward $+ 0.25 \times$ action_reward, yielding an RTS of 5.803. The speed reward and deviation reward motivate the humanoid to run fast, while the action reward promotes smoother motion. In the second iteration, the weight of the speed reward was increased to 0.6, while the weights for deviation and action rewards were adjusted to 0.2 each, improving the RTS to 6.113. In the third iteration, the action penalty was raised and the reward weights were further modified to $0.7 \times$ speed_reward, $0.15 \times$ deviation_reward, and $0.15 \times$ action_reward, resulting in an RTS of 7.915. During the fourth iteration, the deviation penalty was reduced to 0.35 and the action penalty was lowered, with the reward weights set to 0.8, 0.1, and 0.1 for speed, deviation, and action rewards, respectively. This

**Environment (obs dim, action dim)**
Task Description
*Task Metric*

**Cartpole (4, 1)**
To balance a pole on a cart so that the pole stays upright
*duration*

**Quadcopter (21, 12)**
To make the quadcopter reach and hover near a fixed position
*-cur_dist*

**FrankaCabinet (23, 9)**
To open the cabinet door
*1 if cabinet_pos > 0.39*

**Anymal (48, 12)**
To make the quadruped follow randomly chosen x, y, and yaw target velocities
*-(linvel_error + angvel_error)*

**BallBalance (48, 12)**
To keep the ball on the table top without falling
*duration*

**Ant (60, 8)**
To make the ant run forward as fast as possible
*cur_dist - prev_dist*

**AllegroHand (88, 16)**
To make the hand spin the object to a target orientation
*number of consecutive successes where current success is 1 if rot_dist < 0.1*

**Humanoid (108, 21)**
To make the humanoid run as fast as possible
*cur_dist - prev_dist*

**ShadowHand (211, 20)**
To make the shadow hand spin the object to a target orientation
*number of consecutive successes where current success is 1 if rot_dist < 0.1*

Table 5: Details of IssacGym Tasks.

| | Cart. | Ball. | Quad. | Anymal | Ant | Human. | Franka | Shadow | Allegro |
|---|---|---|---|---|---|---|---|---|---|
| PrefPPO-49 | **499**(0) | **499**(0) | -1.066(0.16) | -1.861(0.03) | 0.743(0.20) | 0.457(0.09) | 0.0044(0.00) | 0.0746(0.02) | 0.0125(0.003) |
| PEBBLE-49 | **499**(0) | **499**(0) | -1.1904(0.14) | -1.521 | 5.9891 | 0.903 | 0.0453 | 0.2142 | 0.1467 |
| SURF-49 | **499**(0) | **499**(0) | -1.208(0.03) | -1.35 | 0.815 | 1.675 | 0.0039 | 0.15 | 0.1116 |
| PrefPPO-150 | **499**(0) | **499**(0) | -0.959(0.15) | -1.818(0.07) | 0.171(0.05) | 0.607(0.02) | 0.0179(0.01) | 0.0617(0.01) | 0.0153(0.004) |
| PEBBLE-150 | **499**(0) | **499**(0) | -1.059(0.07) | -1.436 | 7.257 | 3.254 | 0.0532 | 0.2369 | 0.2811 |
| SURF-150 | **499**(0) | **499**(0) | -1.114(0.06) | -1.42 | 4.246 | 4.312 | 0.0453 | 0.2096 | 0.2 |
| PrefPPO-1.5k | **499**(0) | **499**(0) | -0.486(0.11) | -1.417(0.21) | 4.458(1.30) | 1.329(0.33) | 0.3248(0.12) | 0.0488(0.01) | 0.0284(0.005) |
| PEBBLE-1.5k | **499**(0) | **499**(0) | -0.529(0.14) | -1.332 | 8.282 | 4.075 | 0.1622 | 0.2416 | 0.2615 |
| SURF-1.5k | **499**(0) | **499**(0) | -0.308(0.06) | -1.278 | 7.921 | 2.999 | 0.2639 | 0.2355 | 0.2283 |
| PrefPPO-15k | **499**(0) | **499**(0) | -0.250(0.06) | -1.357(0.02) | 4.626(0.57) | 1.317(0.34) | 0.0399(0.02) | 0.0468(0.00) | 0.0157(0.003) |
| PEBBLE-15k | **499**(0) | **499**(0) | -0.231(0.04) | -0.730 | 8.543 | 4.074 | 0.6089 | 0.2438 | 0.2401 |
| SURF-15k | **499**(0) | **499**(0) | -0.266(0.02) | -0.760 | 7.859 | 3.2922 | 0.3434 | 0.2145 | 0.2352 |
| ICPL(Ours) | **499**(0) | **499**(0) | **-0.0195**(0.09) | **-0.007**(0.35) | **12.04**(1.69) | **9.227**(0.93) | **0.9999**(0.24) | **13.231**(1.88) | **25.030**(3.721) |
| Eureka | 499(0) | 499(0) | -0.023(0.07) | -0.003(0.38) | 10.86(0.85) | 9.059(0.83) | 0.9999(0.23) | 11.532(1.38) | 25.250(9.583) |

Table 6: The final task score of all methods across different tasks in IssacGym. The values in parentheses represent the standard deviation.

| | Cart. | Ball. | Quad. | Anymal | Ant | Human. | Franka | Shadow | Allegro |
|---|---|---|---|---|---|---|---|---|---|
| ICPL w/o RT | 499(0) | 499(0) | -0.0340(0.05) | -0.387(0.26) | 10.50(0.45) | 8.337(0.60) | 0.9999(0.25) | 10.769(2.30) | 25.641(9.46) |
| ICPL w/o RTD | 499(0) | 499(0) | -0.0216(0.14) | -0.009(0.38) | 10.53(0.39) | 9.419(2.10) | 1.0000(0.18) | 11.633(1.25) | 23.744(8.80) |
| ICPL w/o RTDB | 499(0) | 499(0) | -0.0136(0.03) | -0.014(0.42) | 11.97(0.71) | 8.214(2.88) | 0.5129(0.06) | 13.663(1.83) | 25.386(3.42) |
| OpenLoop | 499(0) | 499(0) | -0.0410(0.32) | -0.016(0.50) | 9.350(2.34) | 8.306(1.63) | 0.9999(0.22) | 9.476(2.44) | 23.876(7.91) |
| ICPL(Ours) | 499(0) | 499(0) | -0.0195(0.09) | -0.007(0.35) | 12.04(1.69) | 9.227(0.93) | 0.9999(0.24) | 13.231(1.88) | 25.030(3.721) |

Table 7: Ablation studies on ICPL modules. The values in parentheses represent the standard deviation.

| | Cart. | Ball. | Quad. | Anymal | Ant | Human. | Franka | Shadow | Allegro |
|---|---|---|---|---|---|---|---|---|---|
| PrefPPO-49 | 499(0) | 499(0) | -1.288(0.04) | -1.833(0.05) | 0.281(0.06) | 0.855(0.24) | 0.0009(0.00) | 0.1178(0.03) | 0.1000(0.024) |
| PrefPPO-150 | 499(0) | 499(0) | -1.288(0.02) | -1.814(0.07) | 0.545(0.16) | 0.546(0.09) | 0.0012(0.00) | 0.0517(0.01) | 0.0544(0.010) |
| PrefPPO-1.5k | 499(0) | 499(0) | -1.292(0.05) | -1.583(0.13) | 2.235(0.63) | 2.480(0.59) | 0.0077(0.00) | 0.0495(0.01) | 0.0667(0.017) |
| PrefPPO-15k | 499(0) | 499(0) | -1.322(0.04) | -1.611(0.12) | 3.694(0.86) | 1.867(0.19) | 0.0066(0.00) | 0.0543(0.01) | 0.1002(0.030) |
| Eureka | 499(0) | 499(0) | -0.023(0.07) | -0.003(0.38) | 10.86(0.85) | 9.059(0.83) | 0.9999(0.23) | 11.532(1.38) | 25.250(9.583) |
| (Ours) | 499(0) | 499(0) | -0.0195(0.09) | -0.007(0.35) | 12.04(1.69) | 9.227(0.93) | 0.9999(0.24) | 13.231(1.88) | 25.030(3.721) |

Table 8: The final task score of all methods across different tasks in IssacGym, where PrefPPO uses sparse rewards as the preference metric for the simulated teacher. The values in parentheses represent the standard deviation.

change led to an RTS of 8.125. Finally, in the fifth iteration, an additional upright reward term was incorporated, with the total reward calculated as $0.7 \times \text{speed\_reward} + 0.1 \times \text{deviation\_reward} + 0.1 \times \text{action\_reward} + 0.1 \times \text{upright\_reward}$. This adjustment produced the highest RTS of 8.232, allowing ICPL to generate reward functions that were more effectively aligned with the task description. Below are the specific reward functions produced at each iteration during one experiment.

---

**Humanoid Task: Reward Function with highest RTS (5.803) of Iteration 1**

```python
def compute_reward(root_states: torch.Tensor, actions: torch.Tensor
    ) -> Tuple[torch.Tensor, Dict[str, torch.Tensor]]:
    velocity = root_states[: , 7:10]
    forward_velocity = velocity[:, 0]
    target_velocity = 5.0
    deviation_penalty = 0.5
    action_penalty = 0.1

    # Measure how fast the humanoid is going
    speed_reward = torch.exp((forward_velocity - target_velocity))

    # Penalize deviation from the target velocity
    deviation = torch.abs(forward_velocity - target_velocity)
    deviation_reward = torch.exp(-deviation_penalty * deviation)

    # Penalize actions, to induce smoother motion
    sum_actions = torch.sum(actions ** 2, dim=1)
    action_reward = torch.exp(-action_penalty * sum_actions)

    rewards = {'speed_reward': speed_reward, 'deviation_reward':
    deviation_reward, 'action_reward': action_reward }

    total_reward = 0.5 * speed_reward + 0.25 * deviation_reward +
    0.25 * action_reward
    total_reward = total_reward.to(deviation.device)

    return total_reward, rewards
```

---

**Humanoid Task: Reward Function with highest RTS (6.113) of Iteration 2**

```python
def compute_reward(root_states: torch.Tensor, actions: torch.Tensor
    ) -> Tuple[torch.Tensor, Dict[str, torch.Tensor]]:
    velocity = root_states[: , 7:10]
    forward_velocity = velocity[:, 0]
    target_velocity = 5.0
    deviation_penalty = 0.5
    action_penalty = 0.1

    # Measure how fast the humanoid is going
    speed_reward = torch.exp((forward_velocity - target_velocity))
```

```python
        # Penalize deviation from the target velocity
        deviation = torch.abs(forward_velocity - target_velocity)
        deviation_reward = torch.exp(-deviation_penalty * deviation)
        # Penalize actions, to induce smoother motion
        sum_actions = torch.sum(actions ** 2, dim=1)
        action_reward = torch.exp(-action_penalty * sum_actions)

        rewards = {'speed_reward': speed_reward, 'deviation_reward':
        deviation_reward, 'action_reward': action_reward }

        # Adjusted weights in total reward calculation
        total_reward = 0.6 * speed_reward + 0.2 * deviation_reward +
        0.2 * action_reward  # Changed weights
        total_reward = total_reward.to(deviation.device)

        return total_reward, rewards
```

Humanoid Task: Reward Function with highest RTS (7.915) of Iteration 3

```python
def compute_reward(root_states: torch.Tensor, actions: torch.Tensor
    ) -> Tuple[torch.Tensor, Dict[str, torch.Tensor]]:
    velocity = root_states[: , 7:10]
    forward_velocity = velocity[:, 0]
    target_velocity = 5.0
    deviation_penalty = 0.5
    action_penalty = 0.15  # Increased action penalty

    # Measure how fast the humanoid is going
    speed_reward = torch.exp((forward_velocity - target_velocity))
    # Penalize deviation from the target velocity
    deviation = torch.abs(forward_velocity - target_velocity)
    deviation_reward = torch.exp(-deviation_penalty * deviation)
    # Penalize actions, to induce smoother motion
    sum_actions = torch.sum(actions ** 2, dim=1)
    action_reward = torch.exp(-action_penalty * sum_actions)

    rewards = {'speed_reward': speed_reward, 'deviation_reward':
    deviation_reward, 'action_reward': action_reward }

    # Adjusted weights in total reward calculation
    total_reward = 0.7 * speed_reward + 0.15 * deviation_reward +
    0.15 * action_reward  # Changed weights
    total_reward = total_reward.to(deviation.device)

    return total_reward, rewards
```

Humanoid Task: Reward Function with highest RTS (8.125) of Iteration 4

```python
def compute_reward(root_states: torch.Tensor, actions: torch.Tensor
    ) -> Tuple[torch.Tensor, Dict[str, torch.Tensor]]:
    velocity = root_states[:, 7:10]
    forward_velocity = velocity[:, 0]
    target_velocity = 5.0
    deviation_penalty = 0.35  # decrease penalty for deviation
    action_penalty = 0.1  # decrease penalty for action

    # Measure how fast the humanoid is going
    speed_reward = torch.exp((forward_velocity - target_velocity))
```

```
    # Penalize deviation from the target velocity
    deviation = torch.abs(forward_velocity - target_velocity)
    deviation_reward = torch.exp(-deviation_penalty * deviation)
    # Penalize actions, to induce smoother motion
    sum_actions = torch.sum(actions ** 2, dim=1)
    action_reward = torch.exp(-action_penalty * sum_actions)

    rewards = {'speed_reward': speed_reward, 'deviation_reward':
    deviation_reward, 'action_reward': action_reward }

    # Adjust the weights in total reward calculation
    total_reward = 0.8 * speed_reward + 0.1 * deviation_reward +
    0.1 * action_reward # Changed weights
    total_reward = total_reward.to(deviation.device)

    return total_reward, rewards
```

**Humanoid Task: Reward Function with highest RTS (8.233) of Iteration 5**

```
def compute_reward(root_states: torch.Tensor, actions: torch.Tensor
    , up_vec: torch.Tensor) -> Tuple[torch.Tensor, Dict[str, torch.
    Tensor]]:
    velocity = root_states[: , 7:10]
    forward_velocity = velocity[:, 0]
    target_velocity = 5.0
    deviation_penalty = 0.35  # Reduced deviation penalty
    action_penalty = 0.1  # Decreased action penalty

    # Measure how fast the humanoid is going
    speed_reward = torch.exp((forward_velocity - target_velocity))
    # Penalize deviation from the target velocity
    deviation = torch.abs(forward_velocity - target_velocity)
    deviation_reward = torch.exp(-deviation_penalty * deviation)
    # Penalize actions, to induce smoother motion
    sum_actions = torch.sum(actions ** 2, dim=1)
    action_reward = torch.exp(-action_penalty * sum_actions)
    # Reward for maintaining an upright position
    upright_penalty = 1.0  # New upright penalty for the humanoid
    upright_reward = torch.exp(-upright_penalty * (1 - up_vec[:,
    2]))  # Added upright reward

    rewards = {'speed_reward': speed_reward, 'deviation_reward':
    deviation_reward, 'action_reward': action_reward, '
    upright_reward': upright_reward }

    # Adjusted weights in total reward calculation
    total_reward = 0.7 * speed_reward + 0.1 * deviation_reward +
    0.1 * action_reward + 0.1 * upright_reward  # Added upright
    reward to total
    total_reward = total_reward.to(deviation.device)

    return total_reward, rewards
```

## A.6    HUMAN-IN-THE-LOOP PREFERENCE

### A.6.1    DEMOGRAPHIC DATA

The participants in the human-in-the-loop preference experiments consisted of 7 individuals aged 19 to 30, including 2 women and 5 men. Their educational backgrounds included 2 undergraduate

students and 5 graduate students. The 20 volunteers recruited to evaluate the performance of different methods were aged 23 to 28, comprising 5 women and 15 men, with 3 undergraduates and 17 graduate students.

### A.6.2   ISAACGYM TASKS

We evaluate human-in-the-loop preference experiments on tasks in IsaacGym, including *Quadcopter, Humanoid, Ant, ShadowHand, and AllegroHand*. In these experiments, volunteers were limited to comparing reward functions based solely on videos showcasing the final policies derived from each reward function.

In the *Quadcopter* task, humans evaluate performance by observing whether the quadcopter moves quickly and efficiently, and whether it stabilizes in the final position. For the *Humanoid* and *Ant* tasks, where the task description is "make the ant/humanoid run as fast as possible," humans estimate speed by comparing the time taken to cover the same distance and assessing the movement posture. However, due to the variability in movement postures and directions, estimating speed can introduce inaccuracies. In the *ShadowHand* and *AllegroHand* tasks, where the goal is "to make the hand spin the object to a target orientation," Humans find it challenging to calculate the precise difference between the current orientation and the target orientation at every moment, even though the target orientation is displayed nearby. Nevertheless, humans still can estimate the duration of effective rotations with the target orientation in the video, thus evaluating the performance of a single spin. Since the target orientation regenerates upon being reached, the frequency of target orientation changes can also aid in facilitating the assessment of evaluating performance.

Due to the lack of precise environmental data, volunteers cannot make absolutely accurate judgments during the experiments. For instance, in the *Humanoid* task, robots may move in varying directions, which can introduce biases in volunteers' assessments of speed. However, volunteers are still able to filter out extremely poor results and select videos with relatively better performance. In most cases, the selected results closely align with those derived from proxy human preferences, enabling effective improvements in task performance.

Below is a specific case from the *Humanoid* task that illustrates the potential errors humans may make during evaluation and the learning process of the reward function under this assumption. The reward task scores (RTS) chosen by the volunteer across five iterations are $4.521, 6.069, 6.814, 6.363, 6.983$.

In the first iteration, the ground-truth task scores of each policy were $0.593, 2.744, 4.520, 0.192, 2.517, 5.937$, although the volunteer was unaware of these scores. Initially, the volunteer eliminated policies 0 and 3, as the robots in those videos primarily exhibited spinning behavior. Subsequently, the volunteer assessed the speed of the remaining robots based on how quickly a specific robot moved out of the field. The volunteer correctly identified that the robots in policies 1 and 4 were slightly slower. However, due to minor differences in the movement directions of the robots in policies 2 and 5, the volunteer mistakenly selected policy 2 as the best option, incorrectly concluding that the robot in policy 2 was faster.

Thus, the reward function selected in iteration 1 consists of several key components: velocity reward, upright reward, force penalty, unnatural pose penalty, and action penalty. These components not only promote faster training, which is the primary objective, but also encourage the maintenance of an upright pose. Additionally, the function penalizes excessive force usage, extreme joint angles, and large action values to foster smoother and more controlled movements.

In subsequent iterations, the volunteer effectively identified reward functions that exhibited relatively better and worse performance outcomes. Adjustments were made to the weights of each component, and specific temperature values were introduced for each. These modifications resulted in a more balanced reward structure, ensuring that critical aspects exert a stronger influence, thereby allowing for greater control over the learning dynamics and improving the agent's performance in achieving the task. Even in Iteration 4, the volunteer did not select the reward function with the highest RTS (6.813) but instead opted for the second-highest reward function (RTS = 6.363). Nevertheless, the reward function exhibited consistent improvement during these iterations.

Here we show the full reward function during the process.

**Humanoid Task: Reward Function chosen by volunteer with RTS (4.521) of Iteration 1**

```python
def compute_reward(
    velocity: torch.Tensor,
    dof_pos: torch.Tensor,
    dof_force_tensor: torch.Tensor,
    actions: torch.Tensor,
    up_vec: torch.Tensor,
    heading_vec: torch.Tensor
) -> Tuple[torch.Tensor, Dict[str, torch.Tensor]]:
    # Reward for forward velocity (run as fast as possible)
    velocity_reward = velocity[:, 0]

    # Encouragement for upright pose (penalize for deviation from
    vertical)
    upright_reward = up_vec[:, 2]

    # Penalize high force usage (energy efficiency)
    force_penalty = torch.sum(torch.abs(dof_force_tensor), dim=1)

    # Penalize unnatural joint positions (for instance, avoid
    extreme angles)
    unnatural_pose_penalty = torch.sum(torch.abs(dof_pos), dim=1)

    # Penalize for large actions (to promote smoother movement)
    action_penalty = torch.sum(torch.abs(actions), dim=1)

    # Normalize the rewards and penalties
    velocity_reward = torch.exp(velocity_reward) - 1
    upright_reward = torch.exp(upright_reward) - 1
    temperature = 1.0
    force_penalty = torch.exp(-force_penalty / temperature)
    unnatural_pose_penalty = torch.exp(-unnatural_pose_penalty /
    temperature)
    action_penalty = torch.exp(-action_penalty / temperature)

    # Combine the rewards and penalties into a single reward
    total_reward = (
        velocity_reward +
        0.5 * upright_reward -
        0.01 * force_penalty -
        0.01 * unnatural_pose_penalty -
        0.01 * action_penalty
    )

    # Return the total reward and each component for analysis
    reward_components = {
        "velocity_reward": velocity_reward,
        "upright_reward": upright_reward,
        "force_penalty": force_penalty,
        "unnatural_pose_penalty": unnatural_pose_penalty,
        "action_penalty": action_penalty
    }

    return total_reward, reward_components
```

**Humanoid Task: Reward Function chosen by volunteer with RTS (6.069) of Iteration 2**

```python
def compute_reward(
    velocity: torch.Tensor,
```

```
      dof_pos: torch.Tensor,
      dof_force_tensor: torch.Tensor,
      actions: torch.Tensor,
      up_vec: torch.Tensor,
      heading_vec: torch.Tensor
) -> Tuple[torch.Tensor, Dict[str, torch.Tensor]]:
    # Reward for forward velocity (run as fast as possible)
    velocity_reward = velocity[:, 0]
    velocity_temperature = 1.2  # increased slightly
    velocity_reward = torch.exp(velocity_reward /
    velocity_temperature) - 1

    # Encouragement for upright pose (penalize for deviation from
    vertical)
    upright_reward = up_vec[:, 2]
    upright_temperature = 0.5  # introduce a specific temperature
    upright_reward = torch.exp(upright_reward / upright_temperature
    ) - 1

    # Penalize high force usage (energy efficiency)
    force_penalty = torch.sum(torch.abs(dof_force_tensor), dim=1)
    force_temperature = 0.1  # decreased to make it more
    significant
    force_penalty = torch.exp(-force_penalty / force_temperature)

    # Penalize unnatural joint positions (for instance, avoid
    extreme angles)
    unnatural_pose_penalty = torch.sum(torch.abs(dof_pos), dim=1)
    pose_temperature = 0.1  # decreased to make it more significant
    unnatural_pose_penalty = torch.exp(-unnatural_pose_penalty /
    pose_temperature)

    # Penalize for large actions (to promote smoother movement)
    action_penalty = torch.sum(torch.abs(actions), dim=1)
    action_temperature = 0.1  # decreased to make it more
    significant
    action_penalty = torch.exp(-action_penalty / action_temperature
    )

    # Combine the rewards and penalties into a single reward
    total_reward = (
        velocity_reward +
        0.5 * upright_reward -
        0.01 * force_penalty -
        0.01 * unnatural_pose_penalty -
        0.01 * action_penalty
    )

    # Return the total reward and each component for analysis
    reward_components = {
        "velocity_reward": velocity_reward,
        "upright_reward": upright_reward,
        "force_penalty": force_penalty,
        "unnatural_pose_penalty": unnatural_pose_penalty,
        "action_penalty": action_penalty
    }

    return total_reward, reward_components
```

**Humanoid Task: Reward Function chosen by volunteer with RTS (6.814) of Iteration 3**

```python
def compute_reward(
    velocity: torch.Tensor,
    dof_pos: torch.Tensor,
    dof_force_tensor: torch.Tensor,
    actions: torch.Tensor,
    up_vec: torch.Tensor,
    heading_vec: torch.Tensor
) -> Tuple[torch.Tensor, Dict[str, torch.Tensor]]:
    # Reward for forward velocity (run as fast as possible)
    velocity_reward = velocity[:, 0]
    velocity_temperature = 1.1  # minor adjustment
    velocity_reward = torch.exp(velocity_reward /
    velocity_temperature) - 1

    # Encouragement for upright pose (penalize for deviation from
    vertical)
    upright_reward = up_vec[:, 2]
    upright_temperature = 0.6  # slight adjustment
    upright_reward = torch.exp(upright_reward / upright_temperature
    ) - 1

    # Penalize high force usage (energy efficiency)
    force_penalty = torch.sum(torch.abs(dof_force_tensor), dim=1)
    force_temperature = 0.15  # increased to try to make it
    effective
    force_penalty = torch.exp(-force_penalty / force_temperature)

    # Penalize unnatural joint positions (for instance, avoid
    extreme angles)
    unnatural_pose_penalty = torch.sum(torch.abs(dof_pos), dim=1)
    pose_temperature = 0.2  # increased to try to make it effective
    unnatural_pose_penalty = torch.exp(-unnatural_pose_penalty /
    pose_temperature)

    # Penalize for large actions (to promote smoother movement)
    action_penalty = torch.sum(torch.abs(actions), dim=1)
    action_temperature = 0.2  # increased to try to make it
    effective
    action_penalty = torch.exp(-action_penalty / action_temperature
    )

    # Combine the rewards and penalties into a single reward
    total_reward = (
        velocity_reward +
        0.5 * upright_reward -
        0.02 * force_penalty -  # increased slightly for more
    impact
        0.02 * unnatural_pose_penalty -  # increased slightly for
    more impact
        0.02 * action_penalty  # increased slightly for more impact
    )

    # Return the total reward and each component for analysis
    reward_components = {
        "velocity_reward": velocity_reward,
        "upright_reward": upright_reward,
        "force_penalty": force_penalty,
        "unnatural_pose_penalty": unnatural_pose_penalty,
        "action_penalty": action_penalty
    }
```

```
1458
1459         return total_reward, reward_components
1460
1461
1462
1463    Humanoid Task: Reward Function chosen by volunteer with RTS (6.363) of Iteration 4
1464
1465    def compute_reward(
1466        velocity: torch.Tensor,
1467        dof_pos: torch.Tensor,
1468        dof_force_tensor: torch.Tensor,
1469        actions: torch.Tensor,
1470        up_vec: torch.Tensor,
1471        heading_vec: torch.Tensor
1472    ) -> Tuple[torch.Tensor, Dict[str, torch.Tensor]]:
1473        # Reward for forward velocity (run as fast as possible)
1474        velocity_reward = velocity[:, 0]
1475        velocity_temperature = 1.05  # slight adjustment to refine the
1476        impact
1477        velocity_reward = torch.exp(velocity_reward /
1478        velocity_temperature) - 1
1479
1480        # Encouragement for upright pose (penalize for deviation from
1481        vertical)
1482        upright_reward = up_vec[:, 2]
1483        upright_temperature = 0.65  # slight loosening for more upright
1484         reward
1485        upright_reward = torch.exp(upright_reward / upright_temperature
1486        ) - 1
1487
1488        # Penalize high force usage (energy efficiency)
1489        force_penalty = torch.sum(torch.abs(dof_force_tensor), dim=1)
1490        force_temperature = 0.2  # increased to make it more
1491        significant
1492        force_penalty = torch.exp(-force_penalty / force_temperature)
1493
1494        # Penalize unnatural joint positions (for instance, avoid
1495        extreme angles)
1496        unnatural_pose_penalty = torch.sum(torch.abs(dof_pos), dim=1)
1497        pose_temperature = 0.25  # slight increase to make this
1498        component effective
1499        unnatural_pose_penalty = torch.exp(-unnatural_pose_penalty /
1500        pose_temperature)
1501
1502        # Penalize for large actions (to promote smoother movement)
1503        action_penalty = torch.sum(torch.abs(actions), dim=1)
1504        action_temperature = 0.25  # slightly adjusted for more
1505        prominent constraint
1506        action_penalty = torch.exp(-action_penalty / action_temperature
1507        )
1508
1509        # Combine the rewards and penalties into a single reward
1510        total_reward = (
1511            velocity_reward +
            0.5 * upright_reward -
            0.015 * force_penalty -  # slight increase for more impact
            0.015 * unnatural_pose_penalty -  # slight increase for
        more impact
            0.015 * action_penalty  # slight increase for more impact
        )

        # Return the total reward and each component for analysis
        reward_components = {
```

```
        "velocity_reward": velocity_reward,
        "upright_reward": upright_reward,
        "force_penalty": force_penalty,
        "unnatural_pose_penalty": unnatural_pose_penalty,
        "action_penalty": action_penalty
    }

    return total_reward, reward_components
```

Humanoid Task: Reward Function with best RTS (6.813) of Iteration 4(not chosen by volunteer)

```
def compute_reward(
    velocity: torch.Tensor,
    dof_pos: torch.Tensor,
    dof_force_tensor: torch.Tensor,
    actions: torch.Tensor,
    up_vec: torch.Tensor,
    heading_vec: torch.Tensor
) -> Tuple[torch.Tensor, Dict[str, torch.Tensor]]:
    # Reward for forward velocity (run as fast as possible)
    velocity_reward = velocity[:, 0]
    velocity_temperature = 1.15
    velocity_reward = torch.exp(velocity_reward /
    velocity_temperature) - 1

    # Encouragement for upright pose (penalize for deviation from
    vertical)
    upright_reward = up_vec[:, 2]
    upright_temperature = 0.55
    upright_reward = torch.exp(upright_reward / upright_temperature
    ) - 1

    # Penalize high force usage (energy efficiency)
    force_penalty = torch.sum(torch.abs(dof_force_tensor), dim=1)
    force_temperature = 0.12
    force_penalty = torch.exp(-force_penalty / force_temperature)

    # Penalize unnatural joint positions (for instance, avoid
    extreme angles)
    unnatural_pose_penalty = torch.sum(torch.abs(dof_pos), dim=1)
    pose_temperature = 0.18
    unnatural_pose_penalty = torch.exp(-unnatural_pose_penalty /
    pose_temperature)

    # Penalize for large actions (to promote smoother movement)
    action_penalty = torch.sum(torch.abs(actions), dim=1)
    action_temperature = 0.18
    action_penalty = torch.exp(-action_penalty / action_temperature
    )

    # Combine the rewards and penalties into a single reward
    total_reward = (
        velocity_reward +
        0.5 * upright_reward -
        0.02 * force_penalty -
        0.02 * unnatural_pose_penalty -
        0.02 * action_penalty
    )
```

```
1566
1567        # Return the total reward and each component for analysis
1568        reward_components = {
1569            "velocity_reward": velocity_reward,
1570            "upright_reward": upright_reward,
1571            "force_penalty": force_penalty,
1572            "unnatural_pose_penalty": unnatural_pose_penalty,
1573            "action_penalty": action_penalty
1574        }
1575
1576        return total_reward, reward_components
```

**Humanoid Task: Reward Function chosen by volunteer with RTS (6.983) of Iteration 5**

```
def compute_reward(
    velocity: torch.Tensor,
    dof_pos: torch.Tensor,
    dof_force_tensor: torch.Tensor,
    actions: torch.Tensor,
    up_vec: torch.Tensor,
    heading_vec: torch.Tensor
) -> Tuple[torch.Tensor, Dict[str, torch.Tensor]]:
    # Adjusted parameters based on analysis
    velocity_temperature = 1.1
    upright_temperature = 0.65
    force_temperature = 0.25
    pose_temperature = 0.3
    action_temperature = 0.3

    # Reward for forward velocity (run as fast as possible)
    velocity_reward = velocity[:, 0]
    velocity_reward = torch.exp(velocity_reward /
    velocity_temperature) - 1

    # Encouragement for upright pose (penalize for deviation from
    vertical)
    upright_reward = up_vec[:, 2]
    upright_reward = torch.exp(upright_reward / upright_temperature
    ) - 1

    # Penalize high force usage (energy efficiency)
    force_penalty = torch.sum(torch.abs(dof_force_tensor), dim=1)
    force_penalty = torch.exp(-force_penalty / force_temperature)

    # Penalize unnatural joint positions (for instance, avoid
    extreme angles)
    unnatural_pose_penalty = torch.sum(torch.abs(dof_pos), dim=1)
    unnatural_pose_penalty = torch.exp(-unnatural_pose_penalty /
    pose_temperature)

    # Penalize for large actions (to promote smoother movement)
    action_penalty = torch.sum(torch.abs(actions), dim=1)
    action_penalty = torch.exp(-action_penalty / action_temperature
    )

    # Combine the rewards and penalties into a single reward
    total_reward = (
        velocity_reward +
        0.5 * upright_reward -
        0.02 * force_penalty -
        0.02 * unnatural_pose_penalty -
```

```
        0.02 * action_penalty
    )

    # Return the total reward and each component for analysis
    reward_components = {
        "velocity_reward": velocity_reward,
        "upright_reward": upright_reward,
        "force_penalty": force_penalty,
        "unnatural_pose_penalty": unnatural_pose_penalty,
        "action_penalty": action_penalty
    }

    return total_reward, reward_components
```

### A.6.3 HUMANOIDJUMP TASK

In our study, we introduced a novel task: *HumanoidJump*, with the task description being "to make humanoid jump like a real human." The prompt of environment context in this task is shown in Prompt 5.

Prompt 5: Prompts of Environment Context in *HumanoidJump* Task

```
class HumanoidJump(VecTask):
    """Rest of the environment definition omitted."""
    def compute_observations(self):
        self.gym.refresh_dof_state_tensor(self.sim)
        self.gym.refresh_actor_root_state_tensor(self.sim)
        self.gym.refresh_force_sensor_tensor(self.sim)
        self.gym.refresh_dof_force_tensor(self.sim)

        self.obs_buf[:], self.torso_position[:],
        self.prev_torso_position[:], self.velocity_world[:],
        self.angular_velocity_world[:], self.velocity_local[:],
        self.angular_velocity_local[:], self.up_vec[:],
        self.heading_vec[:], self.right_leg_contact_force[:],
        self.left_leg_contact_force[:] = \
            compute_humanoid_jump_observations(
            self.obs_buf, self.root_states, self.torso_position,
            self.inv_start_rot, self.dof_pos, self.dof_vel,
            self.dof_force_tensor, self.dof_limits_lower,
            self.dof_limits_upper, self.dof_vel_scale,
            self.vec_sensor_tensor, self.actions,
            self.dt, self.contact_force_scale,
            self.angular_velocity_scale,
            self.basis_vec0, self.basis_vec1)

def compute_humanoid_jump_observations(obs_buf, root_states, torso_position, inv_start_rot
, dof_pos, dof_vel, dof_force, dof_limits_lower, dof_limits_upper, dof_vel_scale,
 sensor_force_torques, actions, dt, contact_force_scale, angular_velocity_scale,
 basis_vec0, basis_vec1):
    # type: (Tensor, Tensor, Tensor, Tensor, Tensor, Tensor, Tensor, Tensor, Tensor, float
, Tensor, Tensor, float, float, float, Tensor, Tensor) -> Tuple[Tensor, Tensor, Tensor,
 Tensor, Tensor, Tensor, Tensor, Tensor, Tensor, Tensor, Tensor]

    prev_torso_position_new = torso_position.clone()

    torso_position = root_states[:, 0:3]
    torso_rotation = root_states[:, 3:7]
    velocity_world = root_states[:, 7:10]
    angular_velocity_world = root_states[:, 10:13]

    torso_quat, up_proj, up_vec, heading_vec = compute_heading_and_up_vec(
        torso_rotation, inv_start_rot, basis_vec0, basis_vec1, 2)

    velocity_local, angular_velocity_local, roll, pitch, yaw = compute_rot_new(
        torso_quat, velocity_world, angular_velocity_world)

    roll = normalize_angle(roll).unsqueeze(-1)
    yaw = normalize_angle(yaw).unsqueeze(-1)
    dof_pos_scaled = unscale(dof_pos, dof_limits_lower, dof_limits_upper)
    scale_angular_velocity_local = angular_velocity_local * angular_velocity_scale
```

```
1674          obs = torch.cat((root_states[:, 0:3].view(-1, 3), velocity_local,
1675                          scale_angular_velocity_local,
1676                          yaw, roll, up_proj.unsqueeze(-1),
1677                          dof_pos_scaled, dof_vel * dof_vel_scale,
1678                          dof_force * contact_force_scale,
1679                          sensor_force_torques.view(-1, 12) * contact_force_scale,
1680                          actions), dim=-1)
1681
1682          right_leg_contact_force = sensor_force_torques[:, 0:3]
1683          left_leg_contact_force = sensor_force_torques[:, 6:9]
1684
1685          abdomen_y_pos = dof_pos[:, 0]
1686          abdomen_z_pos = dof_pos[:, 1]
1687          abdomen_x_pos = dof_pos[:, 2]
1688          right_hip_x_pos = dof_pos[:, 3]
1689          right_hip_z_pos = dof_pos[:, 4]
1690          right_hip_y_pos = dof_pos[:, 5]
1691          right_knee_pos = dof_pos[:, 6]
1692          right_ankle_x_pos = dof_pos[:, 7]
1693          right_ankle_y_pos = dof_pos[:, 8]
1694          left_hip_x_pos = dof_pos[:, 9]
1695          left_hip_z_pos = dof_pos[:, 10]
1696          left_hip_y_pos = dof_pos[:, 11]
1697          left_knee_pos = dof_pos[:, 12]
1698          left_ankle_x_pos = dof_pos[:, 13]
1699          left_ankle_y_pos = dof_pos[:, 14]
1700          right_shoulder1_pos = dof_pos[:, 15]
              right_shoulder2_pos = dof_pos[:, 16]
              right_elbow_pos = dof_pos[:, 17]
              left_shoulder1_pos = dof_pos[:, 18]
              left_shoulder2_pos = dof_pos[:, 19]
              left_elbow_pos = dof_pos[:, 20]

              right_shoulder1_action = actions[:, 15]
              right_shoulder2_action = actions[:, 16]
              right_elbow_action = actions[:, 17]
              left_shoulder1_action = actions[:, 18]
              left_shoulder2_action = actions[:, 19]
              left_elbow_action = actions[:, 20]

          return obs, torso_position, prev_torso_position_new, velocity_world,
                  angular_velocity_world, velocity_local, scale_angular_velocity_local,
                  up_vec, heading_vec, right_leg_contact_force, left_leg_contact_force
```

**Reward functions.** We show the reward functions in a trial that successfully evolved a human-like jump: bending both legs to jump. Initially, the reward function focused on encouraging vertical movement while penalizing horizontal displacement, high contact force usage, and improper joint movements. Over time, the scaling factors for the rewards and penalties were gradually adjusted by changing the temperature parameters in the exponential scaling. These adjustments aimed to enhance the model's sensitivity to different movement behaviors. For example, the vertical movement reward's temperature was reduced, leading to more precise rewards for positive vertical movements. Similarly, the horizontal displacement penalty was fine-tuned by modifying its temperature across iterations, either decreasing or increasing the penalty's impact on lateral movements. The contact force penalty evolved by decreasing its temperature to penalize excessive force usage more strongly, especially in the later iterations, making the task more sensitive to leg contact forces. Finally, the joint usage reward was refined by adjusting the temperature to either encourage or discourage certain joint behaviors, with more focus on leg extension and contraction patterns. Overall, the changes primarily revolved around adjusting the sensitivity of different components, refining the balance between rewards and penalties to better align the humanoid's behavior with the desired jumping performance.

---

HumanoidJump Task: Reward Function of Iteration 1

```
def compute_reward(torso_position: torch.Tensor,
    prev_torso_position: torch.Tensor, velocity_world: torch.Tensor,
                    right_leg_contact_force: torch.Tensor,
    left_leg_contact_force: torch.Tensor, dof_pos: torch.Tensor) ->
    Tuple[torch.Tensor, Dict[str, torch.Tensor]]:
    # Ensure all tensors are on the same device
```

```
1728
1729      device = torso_position.device
1730
1731      # Compute vertical torso movement reward
1732      vertical_movement = torso_position[:, 2] - prev_torso_position
          [:, 2]
1733      vertical_movement_reward = torch.clamp(vertical_movement, min
1734      =0.0)  # Reward positive vertical movement
1735      vertical_movement_reward = torch.exp(vertical_movement_reward /
1736       0.1)  # Use exponential scaling with temperature
1737
1738      # Compute horizontal displacement penalty
          horizontal_displacement = torch.sum(torch.abs(torso_position[:,
1739       :2] - prev_torso_position[:, :2]), dim=-1)
1740      horizontal_displacement_penalty = torch.exp(-
1741      horizontal_displacement / 0.1)  # Penalize large movements with
          temperature
1742
1743      # Compute leg forces usage reward
1744      contact_force_usage = torch.sum(torch.abs(
1745      right_leg_contact_force) + torch.abs(left_leg_contact_force),
          dim=-1)
1746      contact_force_usage_penalty = torch.exp(-contact_force_usage /
1747      10.0)  # Penalize high contact force usage with temperature
1748
1749      # Compute joint usage reward (encourages proper leg extension
1750      and contraction)
1751      leg_joints_indices = torch.tensor([6, 7, 8, 12, 13, 14], device
          =device)  # Indices of leg joints
1752      leg_joint_usage = torch.mean(dof_pos[:, leg_joints_indices],
1753      dim=-1)
1754      leg_joint_usage_reward = torch.exp(-torch.abs(leg_joint_usage)
1755      / 0.1)  # Encourage movements from neutral position
1756
1757      # Sum all rewards and penalties
          total_reward = vertical_movement_reward +
1758      horizontal_displacement_penalty + contact_force_usage_penalty +
1759      leg_joint_usage_reward
1760
1761      # Create a dictionary for individual reward components
          reward_components = {
1762          'vertical_movement_reward': vertical_movement_reward,
1763          'horizontal_displacement_penalty':
1764      horizontal_displacement_penalty,
1765          'contact_force_usage_penalty': contact_force_usage_penalty,
1766          'leg_joint_usage_reward': leg_joint_usage_reward
1767      }
1768
      return total_reward, reward_components
1769
1770
1771
```

**HumanoidJump Task: Reward Function of Iteration 2**

```
1772
1773
1774  def compute_reward(
1775      torso_position: torch.Tensor,
1776      prev_torso_position: torch.Tensor,
1777      velocity_world: torch.Tensor,
1778      right_leg_contact_force: torch.Tensor,
1779      left_leg_contact_force: torch.Tensor,
      dof_pos: torch.Tensor
1780  ) -> Tuple[torch.Tensor, Dict[str, torch.Tensor]]:
1781      # Ensure all tensors are on the same device
```

```
      device = torso_position.device

      # Compute vertical torso movement reward
      vertical_movement = torso_position[:, 2] - prev_torso_position
      [:, 2]
      vertical_movement_reward = torch.clamp(vertical_movement, min
      =0.0)  # Reward positive vertical movement
      temperature_vertical = 0.05
      vertical_movement_reward = torch.exp(vertical_movement_reward /
       temperature_vertical)  # Use exponential scaling

      # Compute horizontal displacement penalty
      horizontal_displacement = torch.sum(torch.abs(torso_position[:,
       :2] - prev_torso_position[:, :2]), dim=-1)
      temperature_horizontal = 0.1
      horizontal_displacement_penalty = torch.exp(-
      horizontal_displacement / temperature_horizontal)  # Penalize
      large movements with temperature

      # Compute leg forces usage reward
      contact_force_usage = torch.sum(torch.abs(
      right_leg_contact_force) + torch.abs(left_leg_contact_force),
      dim=-1)
      temperature_force = 1.0
      contact_force_usage_penalty = torch.exp(-contact_force_usage /
      temperature_force)  # Penalize high contact force usage with
      temperature

      # Compute joint usage reward (encourages proper leg extension
      and contraction)
      leg_joints_indices = torch.tensor([6, 7, 8, 12, 13, 14], device
      =device)  # Indices of leg joints
      leg_joint_usage = torch.mean(dof_pos[:, leg_joints_indices],
      dim=-1)
      temperature_joints = 0.05
      leg_joint_usage_reward = torch.exp(-torch.abs(leg_joint_usage)
      / temperature_joints)  # Encourage movements from the neutral
      position

      # Sum all rewards and penalties
      total_reward = vertical_movement_reward +
      horizontal_displacement_penalty + contact_force_usage_penalty +
      leg_joint_usage_reward

      # Create a dictionary for individual reward components
      reward_components = {
          'vertical_movement_reward': vertical_movement_reward,
          'horizontal_displacement_penalty':
      horizontal_displacement_penalty,
          'contact_force_usage_penalty': contact_force_usage_penalty,
          'leg_joint_usage_reward': leg_joint_usage_reward
      }

      return total_reward, reward_components
```

HumanoidJump Task: Reward Function of Iteration 3

```
def compute_reward(
      torso_position: torch.Tensor,
      prev_torso_position: torch.Tensor,
```

```
      velocity_world: torch.Tensor,
      right_leg_contact_force: torch.Tensor,
      left_leg_contact_force: torch.Tensor,
      dof_pos: torch.Tensor
) -> Tuple[torch.Tensor, Dict[str, torch.Tensor]]:
      # Ensure all tensors are on the same device
      device = torso_position.device

      # Compute vertical torso movement reward
      vertical_movement = torso_position[:, 2] - prev_torso_position
      [:, 2]
      vertical_movement_reward = torch.clamp(vertical_movement, min
      =0.0)  # Reward positive vertical movement
      temperature_vertical = 0.05
      vertical_movement_reward = torch.exp(vertical_movement_reward /
       temperature_vertical)  # Use exponential scaling

      # Compute horizontal displacement penalty
      horizontal_displacement = torch.sum(torch.abs(torso_position[:,
       :2] - prev_torso_position[:, :2]), dim=-1)
      temperature_horizontal = 0.05  # Adjusted temperature for finer
       tuning
      horizontal_displacement_penalty = torch.exp(-
      horizontal_displacement / temperature_horizontal)  # Penalize
      large movements

      # Compute leg forces usage reward
      contact_force_usage = torch.sum(torch.abs(
      right_leg_contact_force) + torch.abs(left_leg_contact_force),
      dim=-1)
      temperature_force = 5.0  # Adjusted to make contact force usage
       more noticeable
      contact_force_usage_penalty = torch.exp(-contact_force_usage /
      temperature_force)  # Penalize high contact force usage

      # Compute joint usage reward (encourages proper leg extension
      and contraction)
      leg_joints_indices = torch.tensor([6, 7, 8, 12, 13, 14], device
      =device)  # Indices of leg joints
      leg_joint_usage = torch.mean(dof_pos[:, leg_joints_indices],
      dim=-1)
      temperature_joints = 0.05
      leg_joint_usage_reward = torch.exp(-torch.abs(leg_joint_usage)
      / temperature_joints)  # Encourage movements from the neutral
      position

      # Sum all rewards and penalties
      total_reward = vertical_movement_reward +
      horizontal_displacement_penalty + contact_force_usage_penalty +
      leg_joint_usage_reward

      # Create a dictionary for individual reward components
      reward_components = {
          'vertical_movement_reward': vertical_movement_reward,
          'horizontal_displacement_penalty':
      horizontal_displacement_penalty,
          'contact_force_usage_penalty': contact_force_usage_penalty,
          'leg_joint_usage_reward': leg_joint_usage_reward
      }

      return total_reward, reward_components
```

**HumanoidJump Task: Reward Function of Iteration 4**

```python
def compute_reward(
    torso_position: torch.Tensor,
    prev_torso_position: torch.Tensor,
    velocity_world: torch.Tensor,
    right_leg_contact_force: torch.Tensor,
    left_leg_contact_force: torch.Tensor,
    dof_pos: torch.Tensor
) -> Tuple[torch.Tensor, Dict[str, torch.Tensor]]:
    # Ensure all tensors are on the same device
    device = torso_position.device

    # Compute vertical torso movement reward
    vertical_movement = torso_position[:, 2] - prev_torso_position
    [:, 2]
    vertical_movement_reward = torch.clamp(vertical_movement, min
    =0.0)  # Reward positive vertical movement
    temperature_vertical = 0.04  # Adjusted temperature for better
    sensitivity
    vertical_movement_reward = torch.exp(vertical_movement_reward /
     temperature_vertical)  # Use exponential scaling

    # Compute horizontal displacement penalty
    horizontal_displacement = torch.sum(torch.abs(torso_position[:,
     :2] - prev_torso_position[:, :2]), dim=-1)
    temperature_horizontal = 0.1  # Increased temperature to
    penalize horizontal movement more
    horizontal_displacement_penalty = torch.exp(-
    horizontal_displacement / temperature_horizontal)  # Penalize
    large movements

    # Compute leg forces usage reward
    contact_force_usage = torch.sum(torch.abs(
    right_leg_contact_force) + torch.abs(left_leg_contact_force),
    dim=-1)
    temperature_force = 0.1  # Significantly increase sensitivity
    to contact forces
    contact_force_usage_penalty = torch.exp(-contact_force_usage /
    temperature_force)  # Penalize high contact force usage

    # Compute joint usage reward (encourages proper leg extension
    and contraction)
    leg_joints_indices = torch.tensor([6, 7, 8, 12, 13, 14], device
    =device)  # Indices of leg joints
    leg_joint_usage = torch.mean(dof_pos[:, leg_joints_indices],
    dim=-1)
    temperature_joints = 0.02  # Adjusted for joint usage
    sensitivity
    leg_joint_usage_reward = torch.exp(-torch.abs(leg_joint_usage)
    / temperature_joints)  # Encourage movements from the neutral
    position

    # Sum all rewards and penalties
    total_reward = vertical_movement_reward +
    horizontal_displacement_penalty + contact_force_usage_penalty +
    leg_joint_usage_reward

    # Create a dictionary for individual reward components
    reward_components = {
        'vertical_movement_reward': vertical_movement_reward,
        'horizontal_displacement_penalty':
    horizontal_displacement_penalty,
```

```
            'contact_force_usage_penalty': contact_force_usage_penalty,
            'leg_joint_usage_reward': leg_joint_usage_reward
        }

        return total_reward, reward_components
```

**Humanoid Task: Reward Function of Iteration 5**

```python
def compute_reward(
    torso_position: torch.Tensor,
    prev_torso_position: torch.Tensor,
    velocity_world: torch.Tensor,
    right_leg_contact_force: torch.Tensor,
    left_leg_contact_force: torch.Tensor,
    dof_pos: torch.Tensor
) -> Tuple[torch.Tensor, Dict[str, torch.Tensor]]:
    # Ensure all tensors are on the same device
    device = torso_position.device

    # Compute vertical torso movement reward
    vertical_movement = torso_position[:, 2] - prev_torso_position
    [:, 2]
    vertical_movement_reward = torch.clamp(vertical_movement, min
    =0.0)   # Reward positive vertical movement
    temperature_vertical = 0.04   # Adjusted temperature for better
    sensitivity
    vertical_movement_reward = torch.exp(vertical_movement_reward /
     temperature_vertical)   # Use exponential scaling

    # Compute horizontal displacement penalty
    horizontal_displacement = torch.sum(torch.abs(torso_position[:,
     :2] - prev_torso_position[:, :2]), dim=-1)
    temperature_horizontal = 0.05   # Decreased temperature for more
     sensitivity
    horizontal_displacement_penalty = torch.exp(-
    horizontal_displacement / temperature_horizontal)   # Penalize
    large movements

    # Compute leg forces usage penalty (Rewritten to reduce contact
     force)
    contact_force_usage = torch.sum(torch.abs(
    right_leg_contact_force) + torch.abs(left_leg_contact_force),
    dim=-1)
    temperature_force = 0.5   # Adjusted to penalize contact force
    usage
    contact_force_usage_penalty = torch.exp(-contact_force_usage /
    temperature_force)   # Penalize high contact force usage

    # Compute joint usage reward (encourages proper leg extension
    and contraction)
    leg_joints_indices = torch.tensor([6, 7, 8, 12, 13, 14], device
    =device)   # Indices of leg joints
    leg_joint_usage = torch.mean(torch.abs(dof_pos[:,
    leg_joints_indices]), dim=-1)
    temperature_joints = 0.02   # Adjusted for joint usage
    sensitivity
    leg_joint_usage_reward = torch.exp(-leg_joint_usage /
    temperature_joints)   # Encourage movements from the neutral
    position
```

```
      # Sum all rewards and penalties
      total_reward = vertical_movement_reward +
      horizontal_displacement_penalty + contact_force_usage_penalty +
      leg_joint_usage_reward

      # Create a dictionary for individual reward components
      reward_components = {
          'vertical_movement_reward': vertical_movement_reward,
          'horizontal_displacement_penalty':
      horizontal_displacement_penalty,
          'contact_force_usage_penalty': contact_force_usage_penalty,
          'leg_joint_usage_reward': leg_joint_usage_reward
      }

      return total_reward, reward_components
```

Humanoid Task: Reward Function of Iteration 6

```
def compute_reward(
    torso_position: torch.Tensor,
    prev_torso_position: torch.Tensor,
    velocity_world: torch.Tensor,
    right_leg_contact_force: torch.Tensor,
    left_leg_contact_force: torch.Tensor,
    dof_pos: torch.Tensor
) -> Tuple[torch.Tensor, Dict[str, torch.Tensor]]:
    # Ensure all tensors are on the same device
    device = torso_position.device

    # Compute vertical torso movement reward
    vertical_movement = torso_position[:, 2] - prev_torso_position
    [:, 2]
    vertical_movement_reward = torch.clamp(vertical_movement, min
    =0.0)  # Reward positive vertical movement
    temperature_vertical = 0.03  # Fine-tuned temperature for
    better sensitivity
    vertical_movement_reward = torch.exp(vertical_movement_reward /
     temperature_vertical)  # Use exponential scaling

    # Compute horizontal displacement penalty
    horizontal_displacement = torch.sum(torch.abs(torso_position[:,
     :2] - prev_torso_position[:, :2]), dim=-1)
    temperature_horizontal = 0.04  # Decreased temperature for more
     sensitivity
    horizontal_displacement_penalty = torch.exp(-
    horizontal_displacement / temperature_horizontal)  # Penalize
    large movements

    # Compute leg forces usage penalty (encourage minimal contact
    force)
    contact_force_usage = torch.sum(torch.abs(
    right_leg_contact_force) + torch.abs(left_leg_contact_force),
    dim=-1)
    temperature_force = 0.5  # Adjusted to penalize contact force
    usage
    contact_force_usage_penalty = torch.exp(-contact_force_usage /
    temperature_force)  # Penalize high contact force usage

    # Compute joint usage reward (encourages proper leg extension
    and contraction)
```

```
        leg_joints_indices = torch.tensor([6, 7, 8, 12, 13, 14], device
        =device)  # Indices of leg joints
        leg_joint_usage = torch.mean(torch.abs(dof_pos[:,
        leg_joints_indices]), dim=-1)
        temperature_joints = 0.02  # Fine-tuned for joint usage
        sensitivity
        leg_joint_usage_reward = torch.exp(-torch.abs(leg_joint_usage)
        / temperature_joints)  # Encourage movements from the neutral
        position

        # Sum all rewards and penalties
        total_reward = vertical_movement_reward +
        horizontal_displacement_penalty + contact_force_usage_penalty +
        leg_joint_usage_reward

        # Create a dictionary for individual reward components
        reward_components = {
            'vertical_movement_reward': vertical_movement_reward,
            'horizontal_displacement_penalty':
        horizontal_displacement_penalty,
            'contact_force_usage_penalty': contact_force_usage_penalty,
            'leg_joint_usage_reward': leg_joint_usage_reward
        }

        return total_reward, reward_components
```

