# OpenReview forum: "Few-shot In-context Preference Learning using Large Language Models"
_ICLR.cc/2025/Conference — Submitted to ICLR 2025_

### Official Review · Reviewer_9Tx3 · 2024-10-31

**Soundness:** 2
**Presentation:** 3
**Contribution:** 2
**Rating:** 5
**Confidence:** 2

**Summary:**

This paper proposes a reward design method. It uses LLMs to generate reward functions to calculate the reward, and the prompt of the LLM is learned through human feedback of the policy rollouts and other historical information in the loop.

It replaces the implicit reward model in traditional RLHF with an LLM and its output reward function. This enhances the interoperability and capacity of the reward design.

**Strengths:**

It replaces the implicit reward model in traditional RLHF with an LLM and its output reward function. This enhances the interoperability and capacity of the reward design.

**Weaknesses:**

(1) ICPL involves human labor, but does not show any significant gain over Eureka, which doesn't require any human feedback.

(3) For challenging tasks, true human feedback does not work better than proxy human feedback. This undermines the necessity of involving humans.

(2) For challenging tasks, like humanoid jump task, ICPL does not have any solid comparisons with other baselines.

**Questions:**

(1) Why is it necessary to use pair-wise human feedback (a good example and a bad example) if RTS is available? Why not just use all the reward functions with their RTS as prompt (maybe together with other information like reward trace, differences, etc) to generate reward functions?

(2) Could you please explain the counter-intuitive results in Table 2? It seems the more prompt components you remove (from w/o RT, to w/o RTD, to w/o RTDB), the better performance it gets (w/o RT wins 2 tasks, w/o RTD wins 3 tasks, and w/o RTDB wins 4 tasks), but adding all the components back, i.e., ICPL(Ours), it wins all the tasks.

---

> ### Author Response · Authors · 2024-11-14
>
> We thank the reviewer for highlighting that our work enhances the interoperability and capacity of reward design. The main reason for the reviewer to give a rejection decision is uncertainty as to why we need to involve human feedback in this paper. This appears to stem from a **misunderstanding of the problem we aim to tackle**. We would like to first clarify the scope of our research:
>
> **Our focus is on tasks where humans cannot define any clear reward functions, including both dense and sparse rewards.** In these cases, learning a reward model from human feedback becomes a viable solution. Specifically, humans provide preferences on trajectories without the need to explicitly write out a reward function. In other words, our work is positioned within the realm of preference-based learning methods, where **human feedback serves as a protocol for deriving rewards**. The main contribution of our paper is to **significantly enhance the efficiency of human involvement in preference-based learning by leveraging large language models (LLMs).**
>
> We will address the reviewer’s detailed questions in the following sections.
>
> **Q1: ICPL involves human labor, but does not show any significant gain over Eureka, which doesn't require any human feedback.**
>
> As mentioned earlier, involving human feedback is essential for tackling tasks where humans cannot design any clear reward function. EUREKA, however, still relies on human-designed sparse rewards as fitness scores for evolving reward functions. Therefore, EUREKA is **NOT** a true baseline for our work. We have revised the paper to make this clear. We include it solely to demonstrate an approximate upper bound of performance assuming sparse rewards are available. Our goal is **NOT** to outperform EUREKA, but to show that our approach, which relies only on human preferences, can achieve **surprisingly comparable** results.
>
> **Q2: For challenging tasks, true human feedback does not work better than proxy human feedback. This undermines the necessity of involving humans.**
>
> It is true that real human feedback, which includes noise, can not outperform proxy human feedback, which is noise-free. However, this does not undermine the necessity of involving humans in our tasks. **We believe the reviewer’s point is based on the assumption that sparse rewards are accessible. However, our work focuses on tasks where sparse rewards do not exist.**
>
> First, there might also be a misunderstanding for the reviewer regarding the use of proxy human feedback in our experiments. Tasks without any reward or task metric make it difficult to assess method performance, and conducting real human experiments is time-consuming and not easily scalable. Using proxy human feedback is a standard practice for evaluating preference-based learning methods as It allows for rapid and quantitative comparisons across different methods. In this paper, we use sparse rewards as a proxy for perfect human preferences, but this way is actually not feasible for tasks without sparse rewards. Therefore, real human feedback is necessary in these cases.
>
> Besides, the proxy human feedback is a simulated perfect human without any noise and may not fully capture the challenges humans may face in providing preferences,  we further conduct real human feedback experiments to demonstrate the true effectiveness of our approach. In other words, it is expected that true human feedback doesn’t work better than proxy human feedback, the point we want to show is that our method can still work with true human feedback.
>
> Finally, we want to clarify that our goal is **NOT** to show that human preference or preference-style signals are inherently better than directly using sparse rewards. Rather, we are addressing tasks that lack sparse rewards by incorporating human feedback to guide the learning process.
>
>
> **Q3: For challenging tasks, like humanoid jump task, ICPL does not have any solid comparisons with other baselines.**
>
> EUREKA's reliance on sparse rewards makes it unusable as a baseline in tasks where no explicit reward function is defined, as is the case here: we do not have a sparse reward for this task. Thus, a direct comparison with EUREKA is not feasible. Besides, Table 1 shows that PrefPPO, a preference-based learning baseline, struggles to learn effective behaviors in more challenging tasks, even after 15,000 human queries. Given these limitations, it would be impractical to conduct real human experiments on complex tasks like HumanoidJump based on this performance evaluation.

---

> > ### Author Response · Authors · 2024-11-14
> >
> > **Q4: Why is it necessary to use pair-wise human feedback (a good example and a bad example) if RTS is available? Why not just use all the reward functions with their RTS as prompt (maybe together with other information like reward trace, differences, etc) to generate reward functions**
> >
> > Our aim is **NOT** to demonstrate that human preferences or preference-based signals are inherently superior to sparse rewards but rather to address tasks where sparse rewards are unavailable. We include experiments on tasks with accessible RTS purely for rapid, quantitative evaluation, given the challenge of comparing methods on tasks without RTS. In our method, RTS serves solely as a proxy for human feedback in selecting reward functions, and the RTS value is not visible in the prompts used for learning.
> >
> >
> > **Q5: Could you please explain the counter-intuitive results in Table 2? It seems the more prompt components you remove (from w/o RT, to w/o RTD, to w/o RTDB), the better performance it gets (w/o RT wins 2 tasks, w/o RTD wins 3 tasks, and w/o RTDB wins 4 tasks), but adding all the components back, i.e., ICPL(Ours), it wins all the tasks.**
> >
> > The reasons may be twofold: 1. If the prompt lacks sufficient information, reducing the number of modules—or in other words, shortening the prompt—can help the LLM better identify useful information; 2. LLMs themselves exhibit large variance.
> >
> > We hope our response has addressed the reviewer’s concerns and **kindly request a re-evaluation** of our paper, as there may have been a **misunderstanding of the research goal** in the previous review. Should there be any further questions, we would be more than happy to provide additional clarifications.

---

> ### Comment · Reviewer_9Tx3 · 2024-11-15
>
> Thanks for the clarification. It really helps me to understand the position of this paper. However, I would suggest you elaborate the interested problem and the limitation of other methods in the introduction part. For example, it would be better to move some contents in the related work to the introduction and reorganize the introduction and relate work parts. I will increase the rating if the writing has been improved.
>
> In addition, in Line 037, "However, as task complexity increases or tasks are distinct from the training data, the ability of LLMs to directly write a reward function is limited." Do you show this conclusion in your paper or have any reference? Please let the readers know.
>
> Last, I would like to re-summarize the method and contributions of this paper. Correct me if I'm wrong. This paper combines the power of LLMs and human preference to design the reward efficiently. Comparing to zero-shot LLMs, ICPL uses human preference-based feedback to refine the prompt for LLMs to generate reward functions. Comparing to PrefPPO, ICPL replaces the reward model with LLMs generating reward functions.
>
> Based on the aforementioned two major comparisons, I have the following questions.
>
> **1. About the reward model in PrefPPO**
>
> What size of the reward model do you use in PrefPPO baseline? Would it improve the performance if a larger reward model was employed? I'm curious about this because you use GPT-4o, a very large model, as an (implicit) reward model in some sense, which makes it unfair to compare with a relatively small reward model.
>
> **2. About the zero-shot LLMs**
>
> I still have doubts on the necessity of involving human preference. Let's break it down into a few questions.
>
> (a) I don't see a significant improvement of ICPL over the OpenLoop given that the variance is large. Besides, the OpenLoop performance could be improved given a carefully designed prompt, which brings the following question.
>
> (b) What prompt do you use in OpenLoop? Do you use any prompts that enable the LLM to obtain (implicit) preferences from the trajectories? An example prompt can be "Compare the trajectories carefully, choose the best trajectory, and rewrite the corresponding reward function to improve the performance." If this can work, then it is not necessary to use human preference. The LLM itself can achieve a good result.
>
> (c) What would the performance be if OpenLoop was employed in the humanoid jump task? This is a follow up question for my previous Q3. I would suggest you to establish a baseline on this task because this is the real problem your method aims to tackle, where neither dense nor sparse reward is available. I would also suggest to involve quantitive evaluation. For example, you can involve humans to rate the performance and report the results using Elo score.

---

> ### Author Response · Authors · 2024-11-20
>
> We sincerely appreciate the reviewer's suggestion regarding the reorganization of the introduction and related work sections. We have revised the paper to improve clarity and coherence in these parts. Below, we provide detailed responses to the reviewer’s questions:
>
> **Q1: reward model in PrefPPO**
>
> In PrefPPO, the network architecture of the reward model is a 3-layer MLP with 256 hidden sizes . A reasonable concern arises regarding the impact of larger reward model sizes on performance. To address this, we conducted experiments with three configurations:
> Deeper reward model: 5-layer MLP with 256 hidden units and 10-layer MLP with 256 hidden units.
> Wider reward model: 3-layer MLP with 2048 hidden units.
> Deeper and wider reward model: 5-layer MLP with 1024 hidden units.
> These experiments were conducted on the humanoid task, where the objective is to make the humanoid run as fast as possible. The results show that larger reward model sizes achieve comparable performance to the standard configuration but do not yield performance improvements as the size increases. Additionally, we experimented with a much deeper model (10×256), which resulted in worse performance.
>
> |Setting              | Configuration      | Performance |
> |----------------------|--------------------|-------------|
> | **PrefPPO-1500**     | Original           | 2.47        |
> |                      | Deeper 1 (5×256)     | 3.30   |
> |                      | Deeper 2 (10×256)     | 0.63   |
> |                      | Wider (3×2048)     | 2.80   |
> |                      | Deeper & Wider (5×1024) | 2.75  |
>
> **Q2: What prompt do you use in OpenLoop?**
>
> In the OpenLoop experiment, the LLM does not receive any feedback and only uses a basic prompt containing the task description, observations, and reward-writing tips(Prompt 1 and 3 in Appendix). It generates 30 reward functions across 5 iterations (6 samples per iteration). Additionally, we have tried the example prompt proposed by the reviewer at the beginning of this project, however it is challenging to find a suitable representation of trajectories for the task. We attempted to use VLMs to rank videos, but GPT-4v failed to achieve this effectively. Using state-action pairs as trajectory representations is also problematic, as the LLM struggles to identify the relationship between trajectories and behaviors. Moreover, providing all six reward functions along with their corresponding trajectories to the LLM quickly leads to token limitations, making this approach currently impractical.
>
> **Q3: What would the performance be if OpenLoop was employed in the humanoid jump task?**
>
> Thank you for the suggestion. We have additionally conducted OpenLoop experiments on the humanoid jump task and included the results in the revised paper. Specifically, we performed 5 experiments using the OpenLoop method, with each experiment consisting of 6 iterations and 6 samples per iteration. Then one volunteer selected the most preferred video as the final result for OpenLoop. For quantitative evaluation, we recruited 20 volunteers, including graduate and undergraduate students not involved in this research, to indicate their preferences between two videos shown in random order: one generated by ICPL and the other by OpenLoop. The results show that 85% (17/20) of participants preferred the ICPL agent, indicating that ICPL produces more human-aligned behaviors. These findings demonstrate ICPL's significant advantage over OpenLoop in complex tasks and highlight the necessity of incorporating human preferences in such scenarios.
> | Method    | Human Preference |
> |-----------|------------------|
> | OpenLoop  | 3/20         |
> | ICPL      | 17/20           |

---

> > ### Author Response · Authors · 2024-11-22
> >
> > Dear reviewer,
> >
> > As the discussion period is coming to an end, we kindly ask for your engagement with our rebuttal. We have put significant effort into addressing your concerns and would greatly appreciate any further feedback or discussion.
> >
> > Thank you for the time and thoughtful comments so far!

---

> > > ### Author Response · Authors · 2024-11-26
> > > **Thank You and a Kind Reminder**
> > >
> > > Dear reviewer 9Tx3,
> > >
> > > Thank you for your constructive feedback. We sincerely appreciate your detailed comments, which have significantly helped us improve the quality of our paper. Following your suggestions, we have revised the introduction and related work sections and also included further experiments to validate the effectiveness of ICPL.
> > >
> > > As it is the last day for revisions, we wanted to politely ask if our responses and updates sufficiently addressed the concerns. If there are remaining issues or areas for clarification, we would be more than happy to provide further explanations.
> > > If our responses have resolved the concerns, we would like to understand why our paper is still scored below the acceptance threshold. Your insights would be invaluable in helping us identify areas where we could improve further.
> > > Thank you again for your time and thoughtful feedback.

---

> > > > ### Author Response · Authors · 2024-12-01
> > > > **Follow-Up Regarding Reviewer Feedback**
> > > >
> > > > Dear Reviewer,
> > > >
> > > > Thank you once again for your thoughtful and constructive feedback on our work. We appreciate the time and effort you have devoted to reviewing our paper.
> > > >
> > > > As the discussion period is approaching its conclusion, we would like to kindly follow up to ensure that our responses have adequately addressed your concerns. If there are any additional questions or points requiring further clarification, we would be more than happy to provide detailed explanations.
> > > >
> > > > Looking forward to your reply.

---

### Official Review · Reviewer_U5zu · 2024-11-03

**Soundness:** 3
**Presentation:** 3
**Contribution:** 3
**Rating:** 6
**Confidence:** 5

**Summary:**

The paper proposes In-Context Preference Learning (ICPL), a method using LLMs for more efficient preference-based reinforcement learning. ICPL uses LLMs, such as GPT-4, to synthesize reward functions based on environment context and task descriptions. These generated functions are refined through human feedback iteratively. The approach shows significant efficiency gains, requiring fewer preference queries compared to traditional RLHF and achieving comparable or superior performance to methods using ground-truth reward functions. The experiments validate ICPL's performance across various simulated and real human-in-the-loop RL tasks, showcasing robustness in complex, subjective environments.

**Strengths:**

* Demonstrates a substantial reduction in the number of human queries needed for preference-based learning which is sorely needed since human-in-the-loop approaches should ideally just require a handful of preferences.

* It's appreciated that the evaluations of the method is done both in synthetic data and with real humans.

* The paper is well written and the provided method is explained well. Even tho generating reward functions from LLMs is not novel, the way the iteratively make use of human preferences to update their prompt is.

**Weaknesses:**

* As with all works that uses LLMs to generate reward functions from human feedback I question how well it will perform with more complex tasks which is one of the big reason for using human feedback.

* The synthetic experiment uses completely noiseless preferences while the standard in these kind of control environments are typical a noise of let's say 10%. What is the rationale for using noiseless preferences and what would be the effect of noisy preferences for your method?

* While the authors uses B-Pref from Kimin et al for some reason they use only the PPO version even tho the repository is more associated with PEBBLE the SAC version. Why is SAC not used as well?

* 6 participants are very low for a study with humans. Still, it is better than some papers that run their method with just the authors feedback. It would be nice with some more information about the experiment like demographic data as well as discussing the limitation of a smaller sample size when it comes to generalizability.

Minor things:
* You introduce the same abbreviation on multiple occassions.

* To make the related work more complete, there is another paper using LLMs with preferences.
1. Holk, S., Marta, D., & Leite, I. (2024, March). PREDILECT: Preferences Delineated with Zero-Shot Language-based Reasoning in Reinforcement Learning. In Proceedings of the 2024 ACM/IEEE International Conference on Human-Robot Interaction (pp. 259-268).

**Questions:**

* What was the demographic data for the human provided feedback?

* It seems like Eureka has very similar performance to ICPL, what would you say is the benefit of your method compared to Eureka? Eureka seems to have some constraints but it would be nice to show in experimentation or come up with a scenario where it would fail.

* It would be good to justify the length of the paper. For example, what sections do you believe require the additional space? You are of course free to use all the space but the readability of the paper could improve by making it more crisp.

* Why did you not use PEBBLE as a basline given that you made use of BPref? Also, did you consider any other baselines as there are more recent works [1,2,3] (To name a few)? It would be great if you discuss how you determined which baseline to use and if you considered any others.
1. Kim, C., Park, J., Shin, J., Lee, H., Abbeel, P., & Lee, K. (2023). Preference transformer: Modeling human preferences using transformers for rl. arXiv preprint arXiv:2303.00957.
2. Park, J., Seo, Y., Shin, J., Lee, H., Abbeel, P., & Lee, K. (2022). SURF: Semi-supervised reward learning with data augmentation for feedback-efficient preference-based reinforcement learning. arXiv preprint arXiv:2203.10050.
3. Marta, D., Holk, S., Pek, C., Tumova, J., & Leite, I. (2023, October). VARIQuery: VAE Segment-Based Active Learning for Query Selection in Preference-Based Reinforcement Learning. In 2023 IEEE/RSJ International Conference on Intelligent Robots and Systems (IROS) (pp. 7878-7885). IEEE.

I am more than willing to up the score given reasonable answers to these points.

---

> ### Author Response · Authors · 2024-11-14
>
> We thank the reviewer for highlighting the substantial reduction in the number of human queries, and for noting that the evaluation of our method was conducted using both synthetic data and real humans. We also appreciate the advice on making our paper clearer and more comprehensive. We have revised the paper accordingly, with changes highlighted in red, based on the reviewer’s suggestions. Below are the detailed responses to the reviewer’s questions.
>
> **Q1: As with all works that use LLMs to generate reward functions from human feedback I question how well it will perform with more complex tasks which is one of the big reason for using human feedback.**
>
> It is a reasonable concern that our method may not scale to more complex tasks. However, we would like to clarify two points: 1) we do not claim that our approach will definitively scale to more complex tasks, but rather provide evidence suggesting that it might, and 2) the tasks we evaluate are already sufficiently complex that humans find it challenging to define reward functions for some of them (as demonstrated in EUREKA).
>
> **Q2: The synthetic experiment uses completely noiseless preferences while the standard in these kinds of control environments are typical a noise of let's say 10%. What is the rationale for using noiseless preferences and what would be the effect of noisy preferences for your method?**
>
> We use noiseless preferences initially to verify whether our method can effectively work solely with preference signals, allowing us to assess the upper bound of expected performance for all preference-based learning methods. Additionally, this setup enables a comparison with EUREKA, which uses noiseless sparse rewards as fitness scores to enhance reward functions. Subsequently, we conduct real human experiments, incorporating true noisy preferences. As shown in Table 3, noisy preferences (ICPL-real) perform comparably or slightly lower than noiseless preferences (ICPL-proxy) across all five tasks, yet still surpass the performance without any preferences (OpenLoop) in 3 out of 5 tasks.
>
> **Q3: While the authors uses B-Pref from Kimin et al for some reason they use only the PPO version even tho the repository is more associated with PEBBLE the SAC version. Why is SAC not used as well?**
>
> We adopt PPO as IsaacGym tasks are trained by PPO, which is well-suited to high-parallel, GPU-based simulators. Using different training algorithms, such as PPO and SAC, could result in varying basic performance even with the same human-designed reward function, making it difficult to ensure a fair comparison. Furthermore, this way we can ensure that each algorithm has a well-tuned inner loop in which we train the RL agents.
>
> **Q4:  It would be nice with some more information about the experiment like demographic data as well as discussing the limitation of a smaller sample size when it comes to generalizability.**
>
> Thank you for the suggestion. We have revised the paper to incorporate this information. The participants in our study consisted of six individuals aged 19 to 30, including two women and four men, with educational levels ranging from undergraduate students to graduate students and one postdoc. We acknowledge that the small sample size may limit the generalizability of our findings, as it may not capture the full range of human preferences. Expanding the sample size and diversity is a crucial direction for future work to improve the robustness and applicability of our results.
>
> **Q5: It seems like Eureka has very similar performance to ICPL, what would you say is the benefit of your method compared to Eureka**
>
> Our focus is on tasks where humans cannot define any clear reward functions, including both dense and sparse rewards. EUREKA, however, still relies on human-designed sparse rewards as fitness scores for evolving reward functions. Therefore, EUREKA is not a true baseline for our work. We include it solely to demonstrate the upper bound of performance assuming sparse rewards are available. **Our goal is not to outperform EUREKA, but to show that our approach, which relies only on human preferences, can achieve comparable results. We have revised the paper to make this clear.**
>
>
> **Q6: To make the related work more complete, there is another paper using LLMs with preferences.**
>
> Thanks for pointing that and we have revised the paper to add it in the related work section.

---

> ### Author Response · Authors · 2024-11-14
>
> **Q7: Why did you not use PEBBLE as a baseline given that you made use of BPref? Also, did you consider any other baselines as there are more recent works [1,2,3] (To name a few)? It would be great if you discuss how you determined which baseline to use and if you considered any others.**
>
> Thank you for the suggestion. We selected the most commonly used preference-based learning method as our baseline, but we are happy to incorporate additional baselines. In the revised version, we will report the performance of PEBBLE and SURF as baselines, which update reward models during training as ICPL to ensure a fair comparison.
>
> Thank you again for the valuable feedback. We hope our response has addressed your concerns. If you have any further questions, we would be happy to provide additional information.

---

> > ### Author Response · Authors · 2024-11-25
> > **Results from PEBBLE and SURF**
> >
> > Based on the reviewers helpful suggestion, we have added additional preference learning baselines and uploaded them in the new PDF. We have added PEBBLE as suggested and SURF [1] , a version of PEBBLE that adds data augmentation. In both cases, our conclusions are unchanged and the performance of PrefPPO, PEBBLE, and SURF are all similar and sharply below ICPL. We hope that this strengthens our conclusion that ICPL is an effective approach to preference learning.
> >
> > [1] Jongjin Park, Younggyo Seo, Jinwoo Shin, Honglak Lee, Pieter Abbeel, and Kimin Lee. SURF:
> > Semi-supervised reward learning with data augmentation for feedback-efficient preference-based
> > reinforcement learning. In International Conference on Learning Representations, 2022. URL
> > https://openreview.net/forum?id=TfhfZLQ2EJO.

---

> > > ### Author Response · Authors · 2024-11-26
> > > **Thanks for increasing score**
> > >
> > > We deeply appreciate the reviewer’s acknowledgment of our work and the decision to increase the score. Your feedback has been invaluable in refining and improving the quality of our paper. Thanks again for the kind support!

---

### Official Review · Reviewer_CQ3w · 2024-11-03

**Soundness:** 3
**Presentation:** 4
**Contribution:** 3
**Rating:** 8
**Confidence:** 4

**Summary:**

This paper introduces ICPL, a method for iteratively improving reward functions for RL problems with human feedback. The method has LLMs generate reward functions specified by code, trains and executes these rewards, and then ranks the final trajectories with human feedback to then update the reward functions again.

------
After author response, my main concern (motivation above EUREKA) was clarified and my more minor concerns were addressed or clarified as well, so I have increased my rating and think the paper should be accepted. On balance I think it's a really interesting problem that's being tacked and the experiments (esp human experiments) are really interesting and compelling.

**Strengths:**

The idea is generally really interesting and compelling. The idea of having LLMs generate an initial reward function and then iteratively repeat it is really interesting.

The human study was really compelling and thought out. It's really good that this was actually tried and not just assumed it would work with real human feedback.

Paper really well presented, ideas presented very clearly. Motivation clear and compelling.

**Weaknesses:**

I am struggling to figure out what the compelling advantage is of this method over the baseline Eureka. As far as I understood reading the paper, Eureka operated from the same set of assumptions about the environment as did ICPL. And in the non-human experiment performed very similarly. In the related work it says that EUREKA requires humans to give feedback in text, whereas ICPL only requires ranked preferences. During the description in 5.2 it also says that sparse rewards are used to select the best candidate reward function. Does that mean that this is additional assumptions EUREKA needs. There was also not a comparison to EUREKA in the human study. Was that because it would not work without these other assumptions? I think it's possible I'm just misunderstanding here, so if authors could clarify this point it would really help me understand the paper and potentially improve my rating.

It's stated in the intro and conclusion that ICPL surpasses RLHF is efficiency, but RLHF is not mentioned anywhere in the experiments. Is this an experimental finding of the paper, or are authors just saying based on known findings about the efficiency of RLHF. Could a direct comparison be made in the first (non-human) experiments since you don't need actual humans and can thus potentially run more. More clarity on this point would really help.

Based on 5 iterations, I'm not sure that you can make the claim that it will monotonically improve much past that point. Did authors try past 5 (10, 20).

One sort of undiscussed thing here is that, requiring new models to be trained every iteration does mean that loop is pretty slow. Was 5 iterations chosen for that reason (so it wouldn't take multiple days). This should be maybe discussed as a weakness. E.g. for human studies or using humans, doesn't that mean the humans need to wait hours or else get new humans to provide feedback for every iteration?

**Questions:**

Please clarify the points mentioned above, that would really help me make a better decision about the paper. In particular explaining why this method would be better in some way that EUREKA
(Either because ICPL doesn't require some assumption made by EUREKA or it's better in some other way).

Minor:
Why GPT-4o for the human experiment only? I'm not sure how much it matters actually, but found it curious.

---

> ### Author Response · Authors · 2024-11-14
>
> We sincerely appreciate the reviewer for acknowledging our hard work on the real human study and for recognizing that the idea is generally interesting and compelling. We value this positive feedback, which motivates us to further improve our work. Below, we provide detailed responses to the reviewer’s questions:
>
> **Q1: comparison with EUREKA**
>
> We would like to clarify the scope of our research: **our focus is on tasks where humans cannot define any clear reward functions—whether dense or sparse.** In these cases, learning a reward model from human feedback becomes a viable solution, as humans provide preferences on trajectories without explicitly defining a reward function. Our work, therefore, falls within the domain of preference-based learning, where human feedback serves as the protocol for deriving rewards.
>
> **EUREKA and ICPL work under different assumptions.** EUREKA’s primary goal is to test whether LLMs can produce better reward functions than humans by leveraging human-designed sparse rewards as fitness scores to evolve reward functions. In contrast, ICPL is designed for tasks even without available sparse rewards and leverages LLM grounding to accelerate learning reward functions directly from human preferences. For this reason, **EUREKA is NOT a true baseline for our work, as it cannot address tasks lacking sparse rewards.** We include it only to illustrate an approximate performance upper bound assuming sparse rewards were available. Our goal is not to outperform EUREKA but to show that ICPL, which relies solely on human preferences, can achieve comparable results. **We have revised the paper to make this clear.**
>
> In the EUREKA appendix, an additional experiment uses human text feedback to describe behaviors and improve the initial human-designed reward function. In contrast, our approach relies solely on preferences—yielding higher human-involvement efficiency—and does not rely on an initial human-designed reward function. Note also that in our proxy human preference experiments, EUREKA’s results are based on sparse rewards, not human text feedback.
>
>  **Q2: it's stated in the intro and conclusion that ICPL surpasses RLHF in efficiency, but RLHF is not mentioned anywhere in the experiments.**
>
> In this context, RLHF (reinforcement learning from human feedback) refers to the baseline method PrefPPO. PrefPPO, which learns a reward model from human preferences and subsequently uses this learned reward model to train an RL policy, is a preference-based learning method, i.e., a subset of RLHF.
>
> **Q3: Based on 5 iterations, I'm not sure that you can make the claim that it will monotonically improve much past that point. Did authors try past 5 (10, 20).**
>
> Thank you for pointing that out. We have not tested additional iterations, so we have revised the paper to remove this claim. Our intention is simply to highlight the method’s effectiveness in refining reward functions over time.
>
> **Q4: One sort of undiscussed thing here is that, requiring new models to be trained every iteration does mean that loop is pretty slow. Was 5 iterations chosen for that reason (so it wouldn't take multiple days). This should be maybe discussed as a weakness. E.g. for human studies or using humans, doesn't that mean the humans need to wait hours or else get new humans to provide feedback for every iteration?**
>
> Thanks for the suggestion. The overall time-consuming of one experiment is a limitation of our method. We have revised the limitation part to discuss this. This could be resolved by continually training an RL agent under non-stationary reward functions which could make for good future work.
>
> **Q5: Why GPT-4o for the human experiment only?**
>
> Midway through the paper writing process, we found GPT-4o is cheaper. However,  we had already completed most of the proxy human experiments using GPT-4, so we only switched to GPT-4o for the real human experiments.
>
> We hope our response has addressed the reviewer’s concerns. Should there be any additional questions, we would be glad to offer further clarifications.

---

> > ### Author Response · Authors · 2024-11-22
> >
> > Dear reviewer,
> >
> > As the discussion period is coming to an end, we kindly ask for your engagement with our rebuttal. We have put significant effort into addressing your concerns and would greatly appreciate any further feedback or discussion.
> >
> > Thank you for the time and thoughtful comments so far!

---

> > > ### Comment · Reviewer_CQ3w · 2024-11-22
> > >
> > > Thank you for the clarifications about EUREKA, this was super helpful. I have increased my score accordingly and hope that other reviewers who had this concern do as well.

---

> > > > ### Author Response · Authors · 2024-11-22
> > > > **Appreciated - Thanks!**
> > > >
> > > > Thank you for engaging with our rebuttal and the work! We really appreciate it and the kind words about the work.

---

### Official Review · Reviewer_fiB6 · 2024-11-04

**Soundness:** 3
**Presentation:** 4
**Contribution:** 2
**Rating:** 5
**Confidence:** 3

**Summary:**

This paper presents a novel framework called In-Context Preference Learning (ICPL), which automatically generates dense reward functions by utilizing an LLM capable of querying humans for preference data. The authors find that their method greatly outperforms one baseline, PrefPPO, with respect to sample efficiency (PrefPPO requires far more human preference queries) and task performance. The authors also find performance comparable to that of Eureka, a baseline that also utilizes an LLM for generating dense reward functions but relies upon access to ground-truth sparse reward data rather than human preference data. The authors argue, since ICPL does not require access to a ground-truth sparse reward function, it has a clear advantage for tasks that are less well-defined or require human intuition. Additionally, they argue that training with human preferences will enable greater model alignment.

**Strengths:**

I appreciated the fact that this paper took steps to optimize their baseline methods within reason. For instance, for Eureka, the authors continued generating candidate reward functions until the LLM had generated 6 executable ones (to make things fair for comparison against their own method).

The comparison against PrefPPO was strong.

**Weaknesses:**

According to table 1, ICPL performance seems no better than that of Eureka. Furthermore, substituting in the values from table 3, ICPL performance with real human preference queries does not exceed Eureka’s performance on any task except Ant. Since ICPL does not outperform Eureka, ICPL’s benefit relies upon the ease of obtaining human preference queries in comparison with a ground-truth sparse reward function. I’m not convinced that this benefit is significant.

One argument, from the introduction, is an appeal to the success of preference-based training in other domains. I’m not convinced that this success generalizes to the domain of LLM-generated reward functions.

The other core argument in favor of preference-based training is that human insight—expressed through preference queries—can better align agent behavior with human preferences. The authors motivate this through their custom HumanoidJump task, wherein the task is “to make humanoid jump like a real human.” They argue that this is a domain in which designing a sparse reward function would be difficult due to the nuances/subjectivity of mathematically defining jumping “like a real human.” In my mind, the paper largely hinges on this argument, however the authors only offer one case-study as evidence of the efficacy of human preference data in this domain.

I could be convinced otherwise, but I think there would need to be a more thorough analysis of human preferences in comparison with sparse reward functions in order to be certain.

Also, I found section 5.3.2: Evaluation Metric to be very confusing. I wasn’t sure what an “environment instance” was. I also didn’t understand which set of task metric values was used to compute the maximum for the RTS.

**Questions:**

On page 1, I was confused by the phrase “tasks are distinct from the training data.” What does this mean?
Are there any other reasons to account for why human preference data might be preferable to sparse reward functions?
How do you actually generate the 6 reward function candidates? Do you randomly sample from the LLM? If so, how?

---

> ### Author Response · Authors · 2024-11-14
>
> We thank the reviewer for pointing out that we made reasonable efforts to optimize the baseline methods. We also appreciate the feedback on the strong comparison against PrefPPO. Below, we provide detailed responses to the reviewer’s questions:
>
> **Q1: ICPL in comparison with EUREKA and human preference comparison with sparse reward**
>
> The reviewer’s question regarding the comparison between ICPL and EUREKA, as well as the comparison of human preferences with sparse rewards, suggests **there may be a misunderstanding of the paper's scope. Our focus is on tasks where humans cannot define any clear reward signal, including both sparse and shaped rewards for RL training.** Preference-based learning offers a solution by learning a reward model based on human preferences across different trajectories. For tasks where a clear sparse reward can be defined, human preference is not necessary, and a sparse reward is a more conventional learning signal. Therefore, comparing human preference to sparse reward is NOT the goal of our research. Instead, our objective is to explore how to effectively use human preferences to address tasks that lack clear reward signals.
>
> EUREKA, while also using LLMs to generate reward functions, still relies on sparse rewards as a fitness score for evolving reward functions. Thus, **EUREKA is NOT a direct baseline for our work. It serves as an approximate oracle method to demonstrate the upper bound of performance. We have revised the paper to make this clear.** Our aim is not to outperform EUREKA, but to show that our approach, which relies solely on human preferences, can achieve results that are **comparable**.
>
> Lastly, we want to clarify our use of tasks with sparse rewards in the proxy human preference section. This approach is standard practice for evaluating preference-based learning methods, as it allows for a rapid and quantitative comparison across different methods. Tasks without any reward or task metric make it difficult to assess method performance, and conducting real human experiments can be time-consuming and not easily scalable. Using sparse rewards as a proxy for human preferences to automatically run experiments is a practical and effective evaluation strategy.
>
> **Q2: I’m not convinced that this success generalizes to the domain of LLM-generated reward functions.**
>
> We do not fully understand the point being made here and would appreciate clarification if possible! Our paper is a demonstration that a specific type of preference based-learning does in fact work for LLM-generated reward functions.
>
> **Q3: “ however the authors only offer one case-study as evidence of the efficacy of human preference data in this domain.**
>
> First, our goal is **NOT** to demonstrate that human preferences or preference-based signals are inherently better than directly using sparse rewards. Instead, we focus on addressing tasks that lack clear reward signals by incorporating human feedback to guide the learning process.
>
> Additionally, we conducted **real human experiments** on **six** tasks, including five IsaacGym tasks and one humanoid jump task. In IsaacGym experiments, humans were unaware of the true sparse reward, which allows these tasks to also serve as test cases. One key reason for adopting the five IsaacGym tasks is that they can provide quantitative results by using human-designed sparse rewards as task metrics, enabling us to present evidence of ICPL’s efficacy in this domain.
>
> **Q4: Evaluation Metric to be very confusing. I wasn’t sure what an “environment instance” was. I also didn’t understand which set of task metric values was used to compute the maximum for the RTS.**
>
> We appreciate the reviewer for raising the issue of clarity, and we have revised the paper accordingly. The term 'environment instance' refers to the parallel environments used during RL training. Specifically, we collect data from 10 parallel environments, meaning 10 environment instances. The task metric is the average sparse reward across these parallel environments that return sparse rewards. In each iteration of ICPL, we have six RL training runs. For each RL run, the RTS is the maximum task metric value sampled at fixed intervals. TS represents the maximum of the 30 RTS sets (6 RL runs × 5 iterations).

---

> > ### Author Response · Authors · 2024-11-14
> >
> > **Q5: On page 1, I was confused by the phrase tasks are distinct from the training data. What does this mean**
> >
> > An LLM's ability to zero-shot generate correct reward functions may stem from either the task being relatively simple or the model’s exposure to similar data during training, allowing it to partially memorize the task. However, if a task differs significantly from the data the model has seen in its training set, the LLM lacks the necessary knowledge and will struggle to generate accurate reward functions in a zero-shot manner. The initial low performance in Figure 2 and Figure 3 shows that the reward function is likely not memorized and that ICPL is capable of enhancing performance through the iterative incorporation of preferences.
> >
> > **Q6: How do you actually generate the 6 reward function candidates? Do you randomly sample from the LLM? If so, how?**
> >
> > The prompt sent to the LLM includes the task description, environment observations, necessary variables, and guidelines for writing reward functions. The LLM is then asked to generate six reward functions. Each generated function is parsed for its reward signature and then tested for executability. For any reward functions that fail, we regenerate replacements until we obtain six executable reward functions in total.
> >
> > We hope our response has addressed the reviewer’s concerns, and **we kindly request a re-evaluation of our paper**. If there are any further questions, we are more than happy to provide additional clarifications.

---

> > > ### Author Response · Authors · 2024-11-22
> > >
> > > Dear reviewer,
> > >
> > > As the discussion period is coming to an end, we kindly ask for your engagement with our rebuttal. We have put significant effort into addressing your concerns and would greatly appreciate any further feedback or discussion.
> > >
> > > Thank you for the time and thoughtful comments so far!

---

> > > > ### Comment · Reviewer_fiB6 · 2024-11-24
> > > >
> > > > I appreciate the authors' detailed response to my questions!
> > > >
> > > > **Q1: ICPL in comparison with EUREKA and human preference comparison with sparse reward**
> > > > I believe I understand the scope of this paper: the authors don't aim to directly compare human preferences and reward functions (with respect to ease of implementation, performance, etc.). This is why they don't feel that the comparison against Eureka is valid---instead, they isolate their research scope to the use of human preferences, which makes Eureka (in theory) an upper bound for the results they could reasonably expect to achieve.
> > > >
> > > > Overall, I think this is a reasonable scope, especially because I can imagine (many) situations in which defining an explicit reward signal---even simply a sparse one---might be difficult. For this reason, I have increased my original evaluation scores.
> > > >
> > > > That being said, I am still somewhat hung up on the potential triviality with which one could convert back and forth between preferences and sparse rewards. Given that the authors did much of their analysis using sparse rewards as a proxy for human preferences, couldn't the reverse direction be done somewhat easily (i.e., converting preferences to sparse rewards and then simply using Eureka)? If so, then I worry that the overall contribution of this work is somewhat low. If not, then I think the contribution of this work would be relatively high. It would be great to hear the authors' perspectives on this matter.
> > > >
> > > > **Q2: I’m not convinced that this success generalizes to the domain of LLM-generated reward functions.**
> > > > This question/assertion is now moot because I was referencing a portion of the introduction that the authors have since revised. Originally, I was simply arguing that success of preference-based training in some paradigms (e.g., RLHF) doesn't necessarily imply that preference-based training is especially useful for the paradigm of LLM-generated reward functions. This point was subservient to my broader argument in Q1 regarding the utility of preference-based training for using LLMs to generate reward functions. Thus, this is a non-issue as long as Q1 is addressed.
> > > >
> > > > **Q3: "however the authors only offer one case-study as evidence of the efficacy of human preference data in this domain."**
> > > > I appreciate the authors' point that they performed a much larger number of real human preference experiments than simply one case-study. I originally wrote "one case-study" because HumanoidJump is the only domain that the authors explored in which there is no sparse reward. I agree with the authors' argument that there are many situations in which defining a sparse reward would be difficult; HumanoidJump is a very solid demonstration of a domain in which this difficulty applies. I thus don't think this is a problem with the paper as long as the issues in Q1 are addressed.
> > > >
> > > >
> > > > **Q4: Evaluation Metric to be very confusing. I wasn’t sure what an “environment instance” was. I also didn’t understand which set of task metric values was used to compute the maximum for the RTS.**
> > > > Thank you very much for the clarification and for the update to the paper.
> > > >
> > > > **Q5: On page 1, I was confused by the phrase "tasks are distinct from the training data". What does this mean**
> > > > Thank you for the explanation; I understand what the authors were saying now.
> > > >
> > > > **Q6: How do you actually generate the 6 reward function candidates? Do you randomly sample from the LLM? If so, how?**
> > > > Thank you for the explanation. I was primarily trying to ask how you ensure that the LLM-generated responses are sufficiently different from each other. Do you utilize a certain temperature value to introduce randomness?

---

> ### Author Response · Authors · 2024-11-24
> **Thank you and clarification!**
>
> Thank you to the reviewer for engaging and increasing the score. The points you raised have helped us clarify the paper so far. We would like to make one key clarification about EUREKA and we appreciate that the reviewer pointed it out as it suggests that more rewriting of the introduction is necessary which we will do shortly.
>
> **Q1**: The difference between our method and EUREKA is that EUREKA is a reward shaping method and ICPL is a preference learning method. In EUREKA the reward is known but sparse and hard to do RL with. Thus, an LLM is used to learn reward functions that are easy to learn on and are aligned with the sparse reward. In ICPL, just as in preference learning generally, the reward is unknown and we must learn a functional form for it. An LLM is used both to write out the functional form (in this case code) for the reward function. It then iteratively updates the reward through rounds of preference learning by storing prior rewards and preferences in context and then asking for rewards that match the preferences. So, in ICPL we are in fact converting the preferences directly into a reward function, a problem that is extremely challenging! As you can see from our PrefPPO experiments, naive preference learning does not match our constructed reward functions even with orders of magnitude more samples. As the reviewer suggests, we could then do EUREKA on top of this to make it easier for the RL agent to learn the policy corresponding to this reward. However, it turns out that we don't need to since we already achieve EUREKA-level performance on this task!
>
> To summarize:
> - The LLM is iteratively outputting rewards simply from a task description and a set of preferences. In each round, it puts the previous preferences and rewards in context and outputs 6 new rewards. It turns out the rewards constructed this way match the performance of EUREKA, as measured by the sparse rewards constructed in the EUREKA paper, even though we only have a text description of the task and a small number of preferences.
> - EUREKA does reward shaping given a known reward to try to learn a policy that is optimal under the known reward; we attempt to both learn this known reward purely from preferences and output a policy that is optimal under it. Note that the LLM is outputting an explicit reward signal. **We are doing exactly what the reviewer suggested** and converting the preferences into rewards although the rewards are not necessarily sparse.
>
> **Q2-Q3**: Hopefully our response to Q1 addresses this!
>
> **Q6**: Great question. We use a temperature of 1.0 and will add this to the appendix.

---

> > ### Author Response · Authors · 2024-11-26
> > **Thank You and a Kind Reminder**
> >
> > Dear Reviewer fiB6,
> >
> > Thank you for your timely response and for expressing the remaining concern. We have made further clarifications in response to this issue. The key challenge in preference-based learning is how to convert preferences into rewards, whether dense or sparse. The reverse direction proposed by the reviewer—i.e., converting preferences into sparse rewards—is indeed difficult and exactly the problem we are tackling in our work. We firmly believe that our contribution aligns with the reviewer’s statement that “the contribution of this work would be relatively high.”
> >
> > Since it is the last day for revisions, we would like to ensure that we have sufficiently addressed your concern. If there are still any issues or points requiring further clarification, we would be more than happy to provide additional explanations. If our revisions have resolved your concerns, we would appreciate your feedback on why our paper is still scored below the acceptance threshold. Your insights would be invaluable in helping us identify areas where we can further improve.
> >
> > We sincerely thank you for your time and effort in helping us enhance the quality of our paper.

---

> > > ### Author Response · Authors · 2024-12-01
> > > **Follow-Up Regarding Reviewer Feedback**
> > >
> > > Dear Reviewer,
> > >
> > > Thank you once again for your thoughtful and constructive feedback on our work. We appreciate the time and effort you have devoted to reviewing our paper.
> > >
> > > As the discussion period is approaching its conclusion, we would like to kindly follow up to ensure that our responses have adequately addressed your concerns. If there are any additional questions or points requiring further clarification, we would be more than happy to provide detailed explanations.
> > >
> > > Looking forward to your reply.

---

> > > > ### Comment · Reviewer_fiB6 · 2024-12-02
> > > >
> > > > Regarding Q2, Q3, and Q6, thank you very much for addressing these concerns/questions.
> > > >
> > > > With regards to Q1, I appreciate the additional clarifications, but I still feel that ICPL is not a sufficiently large contribution over Eureka's existing capabilities. Even though ICPL enables preference-based learning, which Eureka was not capable of doing, the modifications make to Eureka to enable this capability seem relatively small in my opinion. For this reason, I must keep my score unchanged.

---

> > > > > ### Author Response · Authors · 2024-12-03
> > > > > **Thank You and Clarification!**
> > > > >
> > > > > Dear Reviewer,
> > > > >
> > > > > Thank you for your thoughtful feedback and for taking the time to review our work. We greatly appreciate your insights and would like to offer further clarifications regarding the contributions of our paper.
> > > > >
> > > > > First, we want to emphasize that our paper is not intended as an improvement over Eureka but rather as a demonstration of a novel capability: that LLMs can perform in-context preference learning (ICPL). This is a finding that had not been previously established. Furthermore, we highlight that ICPL achieves in-context preference learning with orders of magnitude better human efficiency than standard preference learning algorithms.
> > > > >
> > > > > Second, while our work and Eureka both leverage LLMs to generate rewards, we respectfully argue that evaluating our work based on “the amount of work needed” or the rough number of lines of code compared to Eureka is not an appropriate criterion for assessing the contribution of our paper. Demonstrating that ICPL works at all required significant effort, including thousands of hours of experiments and dozens of hours of human trials.
> > > > >
> > > > > Given these points, we kindly request the reviewer reconsider the evaluation of our paper in light of its novelty and the substantial effort involved. Once again, we sincerely thank you for the time and effort you have devoted to reviewing our work.

---

### Author Response · Authors · 2024-11-14
**Overall response to all reviewers**

We sincerely thank the reviewers for their valuable feedback, which has significantly enhanced the quality of the paper. We have made revisions based on their suggestions to improve the clarity and rigor of our work. However, we want to draw the attention of all reviewers to a misunderstanding that is consistent across reviews: **a common criticism that our method does not outperform EUREKA**. Our method should ***NOT*** be expected to beat EUREKA as EUREKA **has access to the actual reward instead of just preferences**. We only included EUREKA as an approximate upper bound on the expected performance ICPL could achieve. Rather than being a negative (i.e. we failed to outperform EUREKA), it is instead surprising that our method is able to match it. **We have revised the paper to make this clear.**

Also we want to clarify that our goal is **NOT** to show that human preference or preference-style signals are inherently better than directly using rewards. Rather, we are addressing tasks that lack any rewards by incorporating human feedback to guide the learning process. This has also been revised to be clearer in the paper.

---

### Author Response · Authors · 2024-11-25
**Addition of two new baselines**

Based on the reviewer U5zu's helpful suggestion, we have added two additional preference learning baselines and uploaded them in the new PDF. We have added PEBBLE [1] as they suggested and additionally added SURF [2] , a version of PEBBLE that adds data augmentation. In both cases, our conclusions are unchanged and the performance of PrefPPO, PEBBLE, and SURF are all similar and sharply below ICPL. We hope that this strengthens our conclusion that ICPL is an effective approach to preference learning.

[1] Lee, Kimin, Laura Smith, and Pieter Abbeel. "Pebble: Feedback-efficient interactive reinforcement learning via relabeling experience and unsupervised pre-training." arXiv preprint arXiv:2106.05091 (2021).

[2] Jongjin Park, Younggyo Seo, Jinwoo Shin, Honglak Lee, Pieter Abbeel, and Kimin Lee. SURF:
Semi-supervised reward learning with data augmentation for feedback-efficient preference-based
reinforcement learning. In International Conference on Learning Representations, 2022. URL
https://openreview.net/forum?id=TfhfZLQ2EJO.

---

### Meta-Review · Area_Chair_WfVF · 2024-12-21

**Metareview:**

This paper proposes a novel algorithm for using LLMs to synthesize reward functions for RLHF (focusing on applications to robot control). They build on Eureka, an existing approach, modifying it to handle human preference feedback instead of ground truth sparse rewards. One point of contention is the connection between this work and Eureka. The authors point out that Eureka assumes access to the ground truth reward function as feedback. In principle, this "ground truth reward function" could be implemented in the RLHF setting by querying the human to obtain their reward; however, providing numerical reward values is likely difficult for humans. Thus, the authors modify Eureka to instead work with human preference feedback in the form of rankings instead of numerical rewards. This contribution appears somewhat incremental; indeed, the methodology appears to simply modify the Eureka prompt to accommodate ranking feedback.

**Additional Comments On Reviewer Discussion:**

The largest concern the reviewers collectively raised was regarding the connection to Eureka. Some reviewers improved their score when the authors clarified the distinction, but others remained unconvinced about the significance of the distinction. There were also some concerns from the reviewers about using LLMs as a proxy for human feedback, which may not have been adequately addressed.

---

### Decision · Program_Chairs · 2025-01-22

Reject